# FEATURE-ROBUSTNESS, FLATNESS AND GENERALIZATION ERROR FOR DEEP NEURAL NETWORKS

## ABSTRACT

The performance of deep neural networks is often attributed to their automated, task-related feature construction. It remains an open question, though, why this leads to solutions with good generalization, even in cases where the number of parameters is larger than the number of samples. Back in the 90s, Hochreiter and Schmidhuber observed that flatness of the loss surface around a local minimum correlates with low generalization error. For several flatness measures, this correlation has been empirically validated. However, it has recently been shown that existing measures of flatness cannot theoretically be related to generalization: if a network uses ReLU activations, the network function can be reparameterized without changing its output in such a way that flatness is changed almost arbitrarily. This paper proposes a natural modification of existing flatness measures that results in invariance to reparameterization. The proposed measures imply a robustness of the network to changes in the input and the hidden layers. Connecting this feature robustness to generalization leads to a generalized definition of the representativeness of data . With this, the generalization error of a model trained on representative data can be bounded by its feature robustness which depends on our novel flatness measure.

## 1  INTRODUCTION

Neural networks (NNs) have become the state of the art machine learning approach in many applications. An explanation for their superior performance is attributed to their ability to automatically learn suitable features from data. In supervised learning, these features are learned implicitly through minimizing the **empirical error** $\mathcal{E}_{emp}(f, S) = {}^1/|S| \sum_{(x,y) \in S} \ell(f(x), y)$ for a **training set** $S \subset \mathcal{X} \times \mathcal{Y}$ drawn iid according to a **target distribution** $\mathcal{D} : \mathcal{X} \times \mathcal{Y} \to [0, 1]$, and a **loss function** $\ell : \mathcal{Y} \times \mathcal{Y} \to \mathbb{R}_+$. Here, $f : \mathcal{X} \to \mathcal{Y}$ denotes the function represented by a neural network.

It is an open question why minimizing the empirical error during deep neural network training leads to good generalization, even though in many cases the number of network parameters is higher than the number of training examples. That is, why deep neural networks have a low **generalization error**

$$\mathcal{E}_{gen} = \mathbb{E}_{(x,y) \sim \mathcal{D}} \left[ \ell(f(x), y) \right] - \frac{1}{|S|} \sum_{(x,y) \in S} \ell(f(x), y) \tag{1}$$

which is the difference between expected error on the target distribution $\mathcal{D}$ and the empirical error on a finite dataset $S \subset \mathcal{X} \times \mathcal{Y}$.

It has been proposed that good generalization correlates with flat minima of the non-convex loss surface (Hochreiter & Schmidhuber, 1997; 1995) and this correlation has been empirically validated (Keskar et al., 2016; Novak et al., 2018; Wang et al., 2018). Thus, for deep neural networks trained with stochastic gradient descent (SGD), this could present a (partial) explanation for their generalization performance (Zhang et al., 2016), since minibatch SGD tends to converge to flat local minima (Zhang et al., 2018; Jastrzębski et al., 2017). This idea was elaborated on by Chaudhari et al. (2016) who suggest a new training method that favors flat over sharp minima even at the cost of a slightly higher empirical error – indeed solutions found by this algorithm exhibit better generalization performance. Similarly, Dziugaite & Roy (2017) augment the loss to improve generalization and find that this promotes flat minima. However, as Dinh et al. (2017) remarked, current flatness measures—which are based only on the Hessian of the loss function—cannot

theoretically be related to generalization: For deep neural networks with ReLU activation functions, there are layer-wise reparameterizations that leave the network function unchanged (hence, also the generalization performance), but change any measure derived only from the loss Hessian.

Another, more intuitive explanation for generalization is that the function generalizes well if the extracted features encode a semantic similarity of the input that is robust to small changes—both in the input and the features. This allows to generalize from the training set to novel, sufficiently similar data. Starting from such a concept of robustness with respect to changes of features, we derive a measure of flatness that is invariant under the mentioned reparameterizations and that reduces to the well-known ridge regression penalty in the special case of a linear regression.

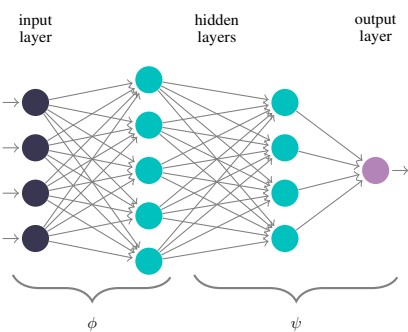

Figure 1: Illustration of the decomposition of $f = \psi \circ \phi$.

This brings three seemingly related properties into our focus: flatness, robustness, and generalization. The exact relationship, however, between flatness of the loss surface around local minima (measuring changes of the empirical error for perturbations in parameter space), robustness (measuring changes of the error for perturbations in either input or feature space), and generalization (performance on unseen data from the target distribution) is not well-understood. This paper provides new insights into this relationship.

The notion of feature robustness proposed in this paper measures the robustness of a function $f = \psi \circ \phi$ (e.g., a neural network) toward local changes in a feature space. That is, $f$ can be split into a composition of functions $f(x) = (\psi \circ \phi)(x)$ for $x \in \mathcal{X}$, $\phi : \mathcal{X} \to \mathbb{R}^m$ and $\psi : \mathbb{R}^m \to \mathcal{Y}$. The function $\phi$ is considered as a feature extraction, mapping the input $\mathcal{X}$ into a **feature space** $\mathbb{R}^m$, while the function $\psi$ corresponds to the model (e.g., a classifier) with $\mathbb{R}^m$ as its domain (see Figure 1 for illustration). It is the feature space defined by $\phi$ where we measure robustness toward small perturbations. For neural networks, the activation values of any but the output layer can be viewed as a feature space. A function $f$ is called **$\epsilon$-feature robust** on a dataset $S \subset \mathcal{X} \times \mathcal{Y}$ if small changes in the feature space defined by $\phi$ do not change the empirical error by more than $\epsilon$. This differs from the notion of robustness defined by Xu & Mannor (2012) using a cover of the sample space, which has been theoretically connected to generalization. Flatness of the loss surface, however, is a local property and we require a more local version of robustness to derive a connection between flatness and robustness. Then, indeed, feature-robustness is upper bounded by the proposed flatness measure. To finally connect the two local properties of robustness and flatness to generalization, we necessarily need a notion describing how representative the given samples are for the true distribution. We define a suitable notion, leading to an upper bound for the generalization error given by feature robustness together with representativeness.

In summary, our contributions are as follows: (i) For models of the form $f(x) = (\psi \circ \phi)(x)$ (e.g. most (deep) neural networks) that split up into a feature extractor $\phi$ and a model $\psi$ on the feature space defined by $\phi$, we define a property of feature robustness that measures the change of the loss function under small perturbations of the features. This property is strongly related to flatness of the loss surface at local minima. (ii) We propose a novel flatness measure. For neural networks with ReLU activation functions, it is invariant under layer-wise reparameterization, addressing a shortcoming of previous measures of flatness. (iii) We define a suitable notion of representativeness of a dataset connecting feature robustness to the generalization error in form of an upper bound. (iv) The proposed flatness measure is empirically shown to strongly correlate with good generalization performance. Thereby, we recover Hessian based quantities as measures of flatness.

## 2 FEATURE ROBUSTNESS

We will define a notion of robustness in feature space $\mathbb{R}^m$ for the model $f = (\psi \circ \phi) : \mathcal{X} \to \mathcal{Y}$, which depends on a small number $\delta > 0$, a training set $S$, and a **feature selection** defined by a matrix $A \in \mathbb{R}^{m \times m}$ of **operator norm** $||A|| \le 1$. In the case of neural networks split into a composition according to Figure 1, traditionally, the activation values $\phi_j(x)$ of neurons are considered as feature

values. The feature value defined by the $j$-th neuron in the feature space $\phi(x) \in \mathbb{R}^m$ can be written as $\phi_j(x) = \langle \phi(x), e_j \rangle$, where $e_j$ denotes the $j$-th unit vector and $\langle \cdot, \cdot \rangle$ the scalar product in $\mathbb{R}^m$. However, it was shown by Szegedy et al. (2013) that, for any other direction $v \in \mathbb{R}^m, ||v|| = 1$, the values $\langle \phi(x), v \rangle = \text{proj}_v \phi(x)$ obtained from the projection $\phi(x)$ onto $v$, can be likewise semantically interpreted as a feature. We can single out the feature defined by $v$ from $\phi(x)$ by multiplication with the projection matrix $E_v = vv^T$. Similarly, multiplication of $\phi(x)$ with a matrix $A$ corresponds to a weighted selection of $rank(A)$-many features in parallel (e.g., projection matrices on d-dimensional subspaces correspond to the selection of $d$ many features). This justifies our terminology considering a matrix $A$ as a feature selection. The same way that, for a sample input $x$, non-activated neurons $\phi_j(x) = 0$ are considered as non-expressed features, we call a selection of features defined by matrix $A$ as **non-expressed** whenever $A\phi(x) = 0$.

We define our notion of **feature robustness**. In words, feature robustness measures the mean change in loss over a dataset under small changes of features in the feature space. Hereby, a matrix $A$ determines which features shall be perturbed. For each sample, the perturbation is linear in the expression of the feature. Thereby, we only perturb features that are relevant for the output for a given sample and leave feature values unchanged that are not expressed (in the sense explained above). With

$$\mathcal{F}(\delta, S, A) := \frac{1}{|S|} \sum_{(x,y) \in S} \left[ \ell(\psi(\phi(x) + \delta A \phi(x)), y) - \ell(f(x), y) \right], \tag{2}$$

the precise definition is given as follows:

**Definition 1.** *Let $\ell : \mathcal{Y} \times \mathcal{Y} \to \mathbb{R}_+$ denote a loss function, $\delta$ and $\epsilon$ two strictly positive (small) real numbers, $S = \{(x_i, y_i) \mid i = 1, \dots, N\} \subseteq \mathcal{X} \times \mathcal{Y}$ a set, and $A \in \mathbb{R}^{m \times m}$ a matrix such that $||A|| \leq 1$. A model $f(x) = (\psi \circ \phi)(x)$, which is a composition of functions $\phi : \mathcal{X} \to \mathbb{R}^m$ and $\psi : \mathbb{R}^m \to \mathcal{Y}$, is called $((\delta, S, A), \epsilon)$-**feature robust**, if $|\mathcal{F}(\delta', S, A)| \leq \epsilon$ for all $|\delta'| \leq \delta$.*
*More generally, if $\mathcal{A} \subset \mathbb{R}^{m \times m}$ denotes a probability space over matrices such that $||A|| \leq 1$ for all $A \in \mathcal{A}$, then we call the model $((\delta, S, \mathcal{A}), \epsilon)$-**feature robust on average over** $\mathcal{A}$, if $\mathbb{E}_{A \sim \mathcal{A}} \left[ |\mathcal{F}(\delta', S, A)| \right] \leq \epsilon$ for all $|\delta'| \leq \delta$.*

We will bound feature robustness at local minima for a dataset $S$ uniformly over all feature selections $A$ and dependent on $\delta$. With our interpretation, this corresponds to an upper bound of the change in loss when perturbing features in feature space $\mathbb{R}^m$. In Appendix C.1 we note how feature robustness is related to noise injection in the layer of consideration, which is known to be related to generalization (An, 1996; Bishop, 1995).

## 3 FEATURE ROBUSTNESS IS CONNECTED TO FLATNESS OF THE LOSS CURVE

Consider a function $f(x, \mathbf{w}) = \psi(\mathbf{w}, \phi(x)) = g(\mathbf{w}\phi(x))$, where $\psi$ is the composition of a twice differentiable function $g : \mathbb{R}^d \to \mathcal{Y}$ and a matrix product with a matrix $\mathbf{w} \in \mathbb{R}^{d \times m}$. As before, $\phi : \mathcal{X} \to \mathbb{R}^m$ can be considered as a feature extractor. We fix a loss function $\ell : \mathcal{Y} \times \mathcal{Y} \to \mathbb{R}_+$ for supervised learning and let $\mathbf{w}_*$ denote a choice of parameters for which the empirical error $\mathcal{E}_{emp}(\mathbf{w}, S) = 1/|S| \sum_{(x,y) \in S} \ell(f(x, \mathbf{w}), y)$, considered as a function on $\mathbf{w}$, is at a local minimum on the training set $S = \{(x_i, y_i) \mid i = 1, \dots, N\}$. In the following, we write $z = \phi(x)$.

For any matrix $A \in \mathbb{R}^{m \times m}$ we have that

$$\psi(\mathbf{w}, z + \delta A z) = g(\mathbf{w}(z + \delta A z)) = g((\mathbf{w} + \delta \mathbf{w} A) z) = \psi(\mathbf{w} + \delta \mathbf{w} A, z). \tag{3}$$

Therefore,

$$\mathcal{F}(\delta, S, A) + \mathcal{E}_{emp}(\mathbf{w}, S) = \frac{1}{|S|} \sum_{(x,y) \in S} \ell(\psi(\mathbf{w}, z + A\delta z), y)$$
$$= \frac{1}{|S|} \sum_{(x,y) \in S} \ell(\psi(\mathbf{w} + \delta \mathbf{w} A, z), y). \tag{4}$$

The latter is the empirical error $\mathcal{E}_{emp}(\mathbf{w} + \delta \mathbf{w} A, S)$ of the model $f$ on the dataset $S$ at parameters $\mathbf{w} + \delta \mathbf{w} A$. If $\delta$ is sufficiently small, then by Taylor expansion of $\mathcal{E}_{emp}(\mathbf{w}, S)$ with respect to

parameters $\mathbf{w}$ around the critical point $\mathbf{w}_*$, we have that

$$
\begin{aligned}
\mathcal{E}_{emp}(\mathbf{w}_* + \delta\mathbf{w}_* A, S) &= \mathcal{E}_{emp}(\mathbf{w}_*, S) + \langle \delta\mathbf{w}_* A, \ \nabla\mathcal{E}_{emp}(\mathbf{w}_*, S) \rangle \\
&+ \frac{1}{2}\langle \delta\mathbf{w}_* A, \ H\mathcal{E}_{emp}(\mathbf{w}_*, S) \cdot (\delta\mathbf{w}_* A) \rangle + \mathcal{O}(\delta^3 ||\mathbf{w}_* A||_F^3) \\
&= \mathcal{E}_{emp}(\mathbf{w}_*, S) + \frac{\delta^2}{2}\langle \mathbf{w}_* A, \ H\mathcal{E}_{emp}(\mathbf{w}_*, S) \cdot (\mathbf{w}_* A) \rangle + \mathcal{O}(\delta^3 ||\mathbf{w}_* A||_F^3)
\end{aligned}
\tag{5}
$$

with $H\mathcal{E}_{emp}(\mathbf{w}_*, S)$ denoting the Hessian of the empirical error with respect to $\mathbf{w}$, $\langle \cdot, \cdot \rangle$ the scalar product with vectorized versions of the parameters and $||\mathbf{w}||_F$ the Frobenius norm of $\mathbf{w}$.

Subtracting $\mathcal{E}_{emp}(\mathbf{w}_*, S)$ from (5), maximizing over matrices $||A|| \leq 1$ and using (4), we get that, for any feature selection $A$, the function (2) defining feature robustness is bounded by

$$
\max_{||A|| \leq 1} \mathcal{F}(\delta, S, A) \leq \frac{\delta^2}{2}||\mathbf{w}_*||_F^2 \ \lambda_{max}^H(\mathbf{w}_*) + \mathcal{O}(\delta^3)
\tag{6}
$$

where $\lambda_{max}^H(\mathbf{w}_*)$ denotes the largest eigenvalue of the Hessian $H\mathcal{E}_{emp}(\mathbf{w}_*, S)$ of the empirical error at $\mathbf{w}_*$. Here we used the identity that $\max_{||x||=1} x^T M x = \lambda_{max}^M$ for any symmetric matrix $M$, and that for matrices of norm $||A|| \leq 1$, we have $||\mathbf{w}_* A||_F \leq ||\mathbf{w}_*||_F$. We show details of the proof of (6) in the appendix. We summarize the connection between feature robustness and flatness in terms of the Hessian in the following theorem.

**Theorem 2.** *Let $\ell : \mathcal{Y} \times \mathcal{Y} \to \mathbb{R}_+$ denote a loss function, $\delta$ a strictly positive (small) real number, $A \in \mathbb{R}^{m \times m}$ a matrix with $||A|| \leq 1$, and let $f(x, \mathbf{w}) = g(\mathbf{w}\phi(x))$ be a model with $g$ an arbitrary twice differentiable function on a matrix product of parameters $\mathbf{w}$ and the image of $x$ under a (feature) function $\phi$. Let $\mathbf{w}_*$ denote a local minimum of the empirical error on a dataset $S$.*
*Then the model $f(\mathbf{w}_*)$ is $((\delta, S, A), \epsilon)$-feature robust for $\epsilon = \frac{\delta^2}{2}||\mathbf{w}_*||_F^2 \ \lambda_{max}^H(\mathbf{w}_*) + \mathcal{O}(\delta^3)$.*

## 4 MEASURES OF FLATNESS OF THE LOSS CURVE

Motivated by the relation of feature robustness with the Hessian $H$, we define a novel measure of flatness. Note that the Hessian is computed with respect to those parameters $\mathbf{w}$ that are applied linearly on the feature space $\phi(\mathcal{X}) \subseteq \mathbb{R}^m$.

**Definition 3.** *Let $\ell : \mathcal{Y} \times \mathcal{Y} \to \mathbb{R}_+$ denote a loss function and $f(x, \mathbf{w}) = g(\mathbf{w}\phi(x))$ be a model with $g : \mathbb{R}^m \to \mathcal{Y}$ an arbitrary twice differentiable function on a matrix product of parameters $\mathbf{w}$ and the image of $x$ under a (feature) function $\phi : \mathcal{X} \to \mathbb{R}^m$. Then $\kappa^\phi(\mathbf{w})$ shall denote a flatness measure of the loss surface defined by*

$$
\kappa^\phi(\mathbf{w}) := ||\mathbf{w}||^2 \cdot \lambda_{max}^H(\mathbf{w}).
\tag{7}
$$

Note that small values of $\kappa^\phi(\mathbf{w})$ indicate flatness and high values indicate sharpness.

**Linear regression with squared loss** In the case of linear regression, $f(x, \mathbf{w}) = \mathbf{w}x \in \mathbb{R}$ ($\mathcal{X} = \mathbb{R}^d$, $g = id$ and $\phi = id$), for any loss function $\ell$, we compute second derivatives with respect to the parameters $\mathbf{w} \in \mathbb{R}^d$ as

$$
\frac{\partial^2 \ell}{\partial w_i \partial w_j} = \frac{\partial^2 \ell}{\partial(f(x, \mathbf{w}))^2} x_i x_j
\tag{8}
$$

If $\ell$ is the squared loss function $\ell(\hat{y}, y) = (\hat{y} - y)^2$, then $\partial^2 \ell / \partial \hat{y}^2 = 2$ and the Hessian is independent of the parameters $\mathbf{w}$. In this case, $\kappa^{id} = c \cdot ||\mathbf{w}||^2$ with a constant $c = 2\lambda_{max}(\sum_{x \in S} xx^t)$ and the measure $\kappa^{id}$ reduces to (a constant multiple of) the well-known Tikhonov (ridge) regression penalty.

**Layers of Neural Networks** We consider neural network functions

$$
f(x) = \mathbf{w}_L \sigma(\dots \sigma(\mathbf{w}_2 \sigma(\mathbf{w}_1 x + b_1) + b_2) \dots) + b_L
\tag{9}
$$

of a neural network of $L$ layers with nonlinear activation function $\sigma$. We hide a possible non-linearity at the output by integrating it in a loss function $\ell$ chosen for neural network training. By letting $\phi^l(x) = \sigma(\mathbf{w}_{l-1}\sigma(\dots \sigma(\mathbf{w}_2 \sigma(\mathbf{w}_1 x + b_1) + b_2) \dots) + b_{l-1})$ denote the output of the composition of the first $l-1$ layers and $g^l(z) = \mathbf{w}_L \sigma(\dots \sigma(z + b_l) \dots) + b_L$ the composition of the activation function

of the $l$-th layer together with the rest of layers, we can write for each layer $l$, $f(x, \mathbf{w}_l) = g^l(\mathbf{w}_l \phi^l(x))$. Using (7) we obtain for each layer of the neural network a measure of flatness at parameter values $\mathbf{w}$:

$$\kappa^l(\mathbf{w}) := ||\mathbf{w}_l||^2 \cdot \lambda_{max}^{H,l}(\mathbf{w}_l) \tag{10}$$

with $\lambda_{max}^{H,l}(\mathbf{w}_l)$ the largest eigenvalue of the Hessian of the empirical error with respect to the parameters of the l-th layer. By Theorem 2, $\kappa^l$ is related to small changes of feature values in layer $l$.

**Corollary 4.** *Let $f$ denote a neural network function of an $L$-layer fully connected neural network. For each layer $l, 1 \le l \le L$ of size $n_l$, let $A \in \mathbb{R}^{n_l \times n_l}$ with $||A|| \le 1$ correspond to feature selections of features in the l-th layer of the neural network. Let $\mathbf{w}_{l*}$ denote weights of the l-th layer at a local minimum of the empirical error.*
*Then the neural network is $((\delta, S, A), \epsilon)$- feature robust in layer $l$ at $\mathbf{w}_*$ for $\epsilon = \frac{\delta^2}{2} \kappa^l(\mathbf{w}_*) + \mathcal{O}(\delta^3)$.*

For an everywhere well-defined Hessian of the loss function, we assumed our network function to be twice differentiable. With the usual adjustments (equations only hold almost everywhere in parameter space), we can also consider neural networks with ReLU activation functions. In this case, Dinh et al. (2017) noted that a linear reparameterization of one layer, $\mathbf{w}_l \to \lambda \mathbf{w}_l$ for $\lambda > 0$, can lead to the same network function by simultaneously multiplying another layer by the inverse of $\lambda$, $\mathbf{w}_k \to 1/\lambda \mathbf{w}_k$, $k \ne l$. Representing the same function, the generalization performance remains unchanged. However, this linear reparameterization changes all common measures of the Hessian of the loss. This constitutes an issue in relating flatness of the loss curve to generalization. We counteract this behavior by the multiplication with $||\mathbf{w}_l||^2$.

**Theorem 5.** *Let $f = f(\mathbf{w}_1, \mathbf{w}_2, \dots, \mathbf{w}_L)$ denote a neural network function parameterized by weights $\mathbf{w}_l$ of the l-th layer. Suppose there are positive numbers $\lambda_1, \dots, \lambda_L$ such that $f(\mathbf{w}_1, \mathbf{w}_2, \dots, \mathbf{w}_L) = f(\lambda_1 \mathbf{w}_1, \lambda_2 \mathbf{w}_2, \dots, \lambda_L \mathbf{w}_L)$ for all $\mathbf{w}_l$. Then, with $\mathbf{w} = (\mathbf{w}_1, \mathbf{w}_2, \dots, \mathbf{w}_L)$ and $\mathbf{w}^\lambda = (\lambda_1 \mathbf{w}_1, \lambda_2 \mathbf{w}_2, \dots, \lambda_L \mathbf{w}_L)$, we have*

$$\kappa^l(\mathbf{w}) = \kappa^l(\mathbf{w}^\lambda) \text{ for all } 1 \le l \le L. \tag{11}$$

We provide a proof in Appendix A.2.

**An Averaging Alternative** Experimental work (Ghorbani et al., 2019) suggests that the spectrum of the Hessian has a lot of small values and only a few large outliers. In this case, our flatness measure serving as an upper bound for feature robustness is governed by the outlier. However, feature robustness for different feature selections is governed by different eigenvalues of the Hessian, according to (5). We therefore consider the trace as an average of the spectrum. We will show that this tracial averaging corresponds to feature robustness on average over all orthogonal feature selection matrices. The following theorem specifies this connection between feature robustness and the unnormalized trace $Tr(H\mathcal{E}_{emp}(\mathbf{w}_*))$ of the empirical error at a local minimum $\mathbf{w}_*$. The details and the proof can be found in Appendix A.3.

**Theorem 6.** *Let $\ell : \mathcal{Y} \times \mathcal{Y} \to \mathbb{R}_+$ denote a loss function, $\delta$ a strictly positive (small) real number, and let $f(x, \mathbf{w}) = g(\mathbf{w}\phi(x))$ be a model with $g$ an arbitrary twice differentiable function on a matrix product of parameters $\mathbf{w} \in \mathbb{R}^{d \times m}$ and the image of $x$ under a (feature) function $\phi$. Let $\mathbf{w}_*$ denote a local minimum of the empirical error on a dataset $S$ and $O_m \subset \mathbb{R}^{m \times m}$ denote the set of orthogonal matrices. Then, (i) for each feature selection matrix $||A|| \le 1$ the model $f(\mathbf{w}_*)$ is $((\delta, S, A), \epsilon)$-feature robust for $\epsilon = \frac{\delta^2}{2} ||\mathbf{w}_*||_F^2 Tr(H\mathcal{E}_{emp}(\mathbf{w}_*)) + \mathcal{O}(\delta^3)$, and (ii) the model $f(\mathbf{w}_*)$ is $((\delta, S, O_m), \epsilon)$-feature robust on average over $O_m$ for $\epsilon = \frac{\delta^2}{2m} ||\mathbf{w}_*||_F^2 Tr(H\mathcal{E}_{emp}(\mathbf{w}_*)) + \mathcal{O}(\delta^3)$.*

We therefore consider the unnormalized trace as a suitable and efficiently computable measure of flatness and define for each layer $l$ of a neural network

$$\kappa_{Tr}^l(\mathbf{w}) := ||\mathbf{w}_l||_F^2 \cdot Tr(H\mathcal{E}_{emp}(\mathbf{w}_l, S)). \tag{12}$$

The same arguments as those used to prove Theorem 5 also show the measure $\kappa_{Tr}^l$ to be independent with respect to the same layer-wise reparameterizations. The analogue of Corollary 4 is as follows.

**Corollary 7.** *Let $f$ denote a neural network function of an $L$-layer fully connected neural network. For each layer $l, 1 \le l \le L$ of size $n_l$, let $O_{n_l} \subset \mathbb{R}^{n_l \times n_l}$ denote the set of orthogonal feature selections in the l-th layer of the neural network. Let $\mathbf{w}_{l*} \in \mathbb{R}^{n_{l+1} \times n_l}$ denote weights of the l-th layer at a local minimum of the empirical error. Then the neural network is $((\delta, S, O_n), \epsilon)$- feature robust in layer $l$ on average over $O_n$ at $\mathbf{w}_*$ for $\epsilon = \frac{\delta^2}{2n_l} \kappa_{Tr}^l(\mathbf{w}_*) + \mathcal{O}(\delta^3)$.*

**Remark 8.** *Other Hessian-based measures have been proposed that are invariant under the given reparameterizations. Liang et al. (2019) consider the Fisher-Rao norm defined by* $||\mathbf{w}^T \hat{H}(\mathbf{w})\mathbf{w}||$ *where* $\hat{H}(\mathbf{w}) = (\nabla_{\mathbf{w}}\ell)(\nabla_{\mathbf{w}}\ell)^T$ *denotes the Gauss-Newton approximation of the loss Hessian. Therefore, the Fisher-Rao norm considers the second partial derivative only into the direction defined by the given weight values* $\mathbf{w}$. *Our measure considers all directions of moving away from a local minimum. In particular, in contrast to these measures, we take the full spectrum of the Hessian into account, which results in a natural measure of flatness around a local minimum. We also came across preprints by Tsuzuku et al. (2019) and Rangamani et al. (2019), which propose a similar measure of flatness. While the first one derives the flatness measure from a PAC Baysian approach, the latter considers the Riemannian metric on the quotient manifold obtained from the equivalence relation given by the refactorization of layers as above.*

## 5 FEATURE ROBUSTNESS AND GENERALIZATION

In this section we aim to study the relation between flatness, feature robustness and the generalization error (defined in (1)). The connection of flat local minima with generalization in PAC Baysian bounds has been considered in several works (Neyshabur et al., 2017; Tsuzuku et al., 2019; Dziugaite & Roy, 2017). Arora et al. (2018) relates noise injection to generalization under the same setting. The work of McAllester (1998; 1999) and Langford & Caruana (2001) initiated the PAC-Baysian approach to generalization, which measures the generalization error (usually for the 0-1 loss) of stochastic classifiers. This leads to bounds on the expected true error over a distribution of models $Q$ in terms of the expectation of the empirical error over $Q$ and the Kullback–Leibler divergence between $Q$ and some prior distribution $P$. For example, Neyshabur et al. (2017) use work by McAllester (2003) to derive an inequality relating the generalization error for the 0-1 loss $\ell$ and expected sharpness $\gamma_\nu := \mathbb{E}_\nu \left[ \mathcal{E}_{emp}(\mathbf{w} + \nu, S) \right] - \mathcal{E}_{emp}(\mathbf{w}, S)$ over distribution $\nu$: With probability $(1 - \epsilon)$ over datasets $S$ of size $N$, the expected error $\mathcal{E}_{\mathbf{w}+\nu} := \mathbb{E}_\nu \left[ \mathbb{E}_{(x,y)\sim\mathcal{D}} \left[ \ell(f(x, \mathbf{w} + \nu), y) \right] \right]$ over the "posterior" distribution $(\mathbf{w} + \nu)$ is bounded by $\mathcal{E}_{emp}(\mathbf{w}, S) + \gamma_\nu + 4\sqrt{\frac{1}{N} \left( \text{KL}(\mathbf{w} + \nu || P) + \ln \frac{2N}{\epsilon} \right)}$. Here, $P$ denotes a "prior" distribution which is fixed before seeing any data and KL denotes the Kullback-Leibler divergence. If we aim to use distributions with local support (as considered in feature robustness and the Taylor expansion relating feature robustness to flatness) with a data-independent prior $P$, the KL term goes to infinity as the distribution $\nu$ becomes increasingly localized. Since feature robustness as a local property is related to generalization (Morcos et al., 2018), we aim to connect the local properties of flatness and feature robustness to generalization of a specific model by following a different approach. Our approach will be independent of the loss function and work in the sample space instead of averages over models in parameter space.

Since feature robustness is a local property in neighborhoods around the points $(x, y) \in S$, to connect feature robustness to generalization we necessarily need an assumption of representativeness of the given data samples $S$. A simple computation shows that

$$
\begin{aligned}
\mathcal{E}_{gen}(f, S) = \mathbb{E}_{A\sim\mathcal{A}} \left[ \mathcal{F}(\delta, S, A) \right] \\
+ \left( \mathbb{E}_{(x,y)\sim\mathcal{D}} \left[ \ell(f(x), y) \right] - \frac{1}{|S|} \sum_{(x_i,y_i)\in S} \mathbb{E}_{A\sim\mathcal{A}} \left[ \ell(\psi(\phi(x_i) + \delta A\phi(x_i)), y_i) \right] \right)
\end{aligned} \tag{13}
$$

The first term is exactly feature robustness on average over a probability distribution $\mathcal{A}$ of feature matrices. For the second term, we accordingly define a notion on datasets $S$ that describes how well the loss on the true distribution can be approximated by certain probability distributions. The distributions we consider are composed of a dataset and (local) probability distributions around its points suitably restricted to local distributions $\lambda_i$ and $\nu_i$ centered around the origin $0$.

**Definition 9.** *Let* $\psi : \mathbb{R}^m \to \mathcal{Y}$ *be a model,* $\ell : \mathcal{Y} \times \mathcal{Y} \to \mathbb{R}_+$ *denote a loss function,* $\epsilon$ *a strictly positive (small) real number, and* $S = \{(x_i, y_i) \mid i = 1, \dots, N\} \subseteq \mathcal{X} \times \mathcal{Y}$ *a set. Let* $\Lambda = (\lambda_i, \nu_i)_{1 \le i \le N}$ *denote a family of pairs of probability distributions on* $\mathbb{R}^m \times \mathcal{Y}$, *where each* $\lambda_i$ *and* $\nu_i$ *have support contained in a neighborhood of the origin* $0$. *(i) The pair* $(S, \Lambda)$ *is called*

**$\epsilon$-representative for $\psi$** (with respect to the loss $\ell$ and distribution $\mathcal{D}$) if $|Rep(S,\Lambda)| \leq \epsilon$, where

$$Rep(S,\Lambda) := \mathbb{E}_{(x,y)\sim\mathcal{D}}\left[\ell(\psi(x),y)\right] - \frac{1}{|S|}\sum_{(x_i,y_i)\in S}\mathbb{E}_{(\xi_x,\xi_y)\sim(\lambda_i\times\nu_i)}\left[\ell(\psi(x_i+\xi_x),y_i+\xi_y)\right].$$

(14)

*(ii) With $\Omega$ a collection of families $\Lambda$ as above and $\mathcal{H}$ a hypothesis space, we say that $S$ is $(\boldsymbol{\epsilon},\boldsymbol{\Omega})$-representative for $\mathcal{H}$ if for all $\psi \in \mathcal{H}$ there is $\Lambda_\psi \in \Omega$ such that $(S,\Lambda_\psi)$ is $\epsilon$-representative for $\psi$.*

Interestingly, we naturally derived a definition of representativeness which is a generalization of classical $\epsilon$-representativeness (see e.g. Definition 4.1 in (Shalev-Shwartz & Ben-David, 2014)), justifying the terminology. Indeed, let $\Lambda_0$ denote the family of probability distributions where each $\lambda_i = \delta_0$ and $\nu_i = \delta_0$ have full weight on the origin. Then $S$ is $(\epsilon,\{\Lambda_0\})$-representative exactly when $S$ is $\epsilon$-representative in the classical sense. Further, if $S$ is $\epsilon$-representative and $S$ is $(\epsilon',\Omega)$-representative for some $\Omega$ containing $\Lambda_0$, then $\epsilon' \leq \epsilon$.

In our setting of a model $f(x) = (\psi \circ \phi)(x)$, which is split up into a feature extractor $\phi$ and a model $\psi$, we consider $(\phi(S),\Lambda)$-representativeness for model $\psi$ and specific choices for $\Lambda = \Lambda_{\delta,\mathcal{A}}$. Here, $\Lambda_{\delta,\mathcal{A}}$ is a family of probability distributions induced by a distribution $\mathcal{A}$ on feature matrices $A$ such that $||A|| \leq \delta$ as follows: We assume that a Borel measure $\mu_A$ is defined by a probability distribution $\mathcal{A}$ on matrices $\mathbb{R}^{m\times m}$. We then define Borel measures $\mu_i$ on $\mathbb{R}^m$ by $\mu_i(C) = \mu_A(\{A \mid A\phi(x_i) \in C\})$ for Borel sets $C \subseteq \mathbb{R}^m$. Then $\lambda_i$ is the probability distribution defined by $\mu_i$. We fix the distributions $\nu_i = \delta_0$ and denote the set containing all families of distributions $(\lambda_i,\nu_i)$ that can be generated this way by $\mathfrak{A}_\delta$. The following result is a direct consequence of Equation 13 and our Definition 9.

**Theorem 10.** *Let $f(x) = (\psi \circ \phi)(x)$ be a model composed of functions $\phi : \mathcal{X} \to \mathbb{R}^m$ and $\psi : \mathbb{R}^m \to \mathcal{Y}$. If $f$ is $((\delta,S,A),\epsilon)$-feature robust for all $||A|| \leq 1$ and $\phi(S)$ is $(\epsilon',\mathfrak{A}_\delta)$-representative for some $\mathcal{H}$ containing $\psi$, then the generalization error of $f$ is bounded by $\mathcal{E}_{gen}(f,S) \leq \epsilon + \epsilon'$.*

Hence, for generalization we need a model that is feature-robust and training data that is sampled densely enough. In the trivial case with $\mathcal{A} = \delta_0$ the distribution with full weight on the 0-matrix, we can choose $\delta = 0$ to obtain $\epsilon = 0$ and $\mathcal{E}_{gen} \leq \epsilon'$. The more feature robust a model is, the larger $\delta$ we can consider to use the flexibility of choosing a nontrivial $\mathcal{A}$ to lower the bound on representativeness and therefore the generalization error. We hope that in future work it will be possible to find suitable distributions $\mathcal{A}$ that lead to computable generalization bounds.

## 6 EMPIRICAL EVALUATION

In this section we empirically validate the practical usefulness of the proposed flatness measure. A correlation between generalization and Hessian-based flatness measures at local minima has been observed previously, but the results of Dinh et al. (2017) questioned the usefulness of these measures. We show that our measure does not only overcome the theoretical issues, but also preserves the strong correlation with the generalization error. Previous works mostly use accuracy of the trained model on the testing dataset (Rangamani et al., 2019; Keskar et al., 2016) for evaluating the generalization properties of the achieved minimum. Nevertheless this does not directly correspond to the theoretical definition of the generalization error (1). For measuring the generalization error, we employ a Monte Carlo approximation of the target distribution defined by the testing dataset and measure the difference between loss value on this approximation and empirical error. In order to track the correlation of the flatness measure to the generalization error, sufficiently different minima should be achieved by training. The most popular technique is to train the model with small and large batch size (Rangamani et al., 2019; Keskar et al., 2016; Novak et al., 2018; Wang et al., 2018), which we also employed.

A neural network (LeNet5 (LeCun et al.)) is trained on CIFAR10 multiple times until convergence with various training setups. This way, we obtain network configurations in multiple local minima. In particular four different initialization schemes were considered (Xavier normal, Kaiming uniform, uniform in $(-0.1,0.1)$, normal with $\mu = 0$ and $\sigma^2 = 0.1$), with four different mini-batch sizes (4, 32, 64, 512) and corresponding learning rates to keep the ration between them equal (0.001, 0.008, 0.02, 0.1) for the standard SGD optimizer. Each of the setups was run for 9 times with different random initializations.

Here the generalization error is the difference between summed error values on test samples multiplied by 5 (since the size of the training set is 5 times larger) and summed error values on the training

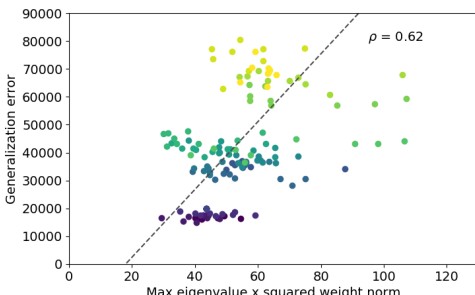 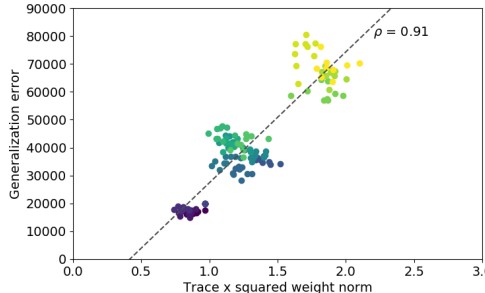

Figure 2: LeNet5 characteristics after training on CIFAR10. Each color corresponds to a different setup of training, characterized by initialization strategy, mini batch size and learning rate. The setups are ordered in ascending order by the mini batch size, with the largest corresponding to the brightest color of the displayed points.

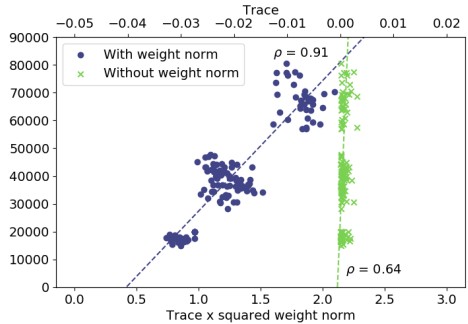 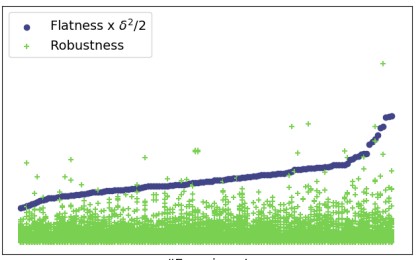

Figure 3: LeNet5 configurations trained on CI-FAR10 with random reparameterizations. The correlation stays the same for the proposed measure, while it breaks for classic Hessian-based measure.

Figure 4: Robustness and flatness for LeNet5 configurations trained on CIFAR10. Results ordered by flatness, showing that robustness is bound by our flatness measure.

examples. Figure 2 shows the approximated generalization error with respect to the flatness measure (for both $\kappa^l$ and $\kappa^l_{Tr}$ with $l = 5$ corresponding to the last hidden layer) for all network configurations. The correlation is significant for both measures, and it is stronger (with $\rho = 0.91$) for $\kappa^5_{Tr}$. This indicates that taking into account the full spectrum of the Hessian is beneficial. To investigate the invariance of the proposed measure to reparameterization, we apply the reparameterization discussed in Sec. 4 to all networks using random factors in the interval $[5, 25]$. The impact of the reparameterization on the proposed flatness measure based on the trace in comparison to the traditional one is shown in Figure 3. While the proposed flatness measure is not affected, the one purely based on the Hessian has very weak correlation with the generalization error after the modifications. To verify the relation described by Equation 6, we also compared feature robustness with $\delta = 0.001$ and feature matrices $A$ that have only one non-zero value $1$ on the diagonal. Figure 4 shows that up to outliers the robustness is bound by the flatness measure. Additional experiments conducted on MNIST dataset are described in Appendix E, where we obtain correlation factors between the generalization error and tracial flatness $\kappa^l_{Tr}$ of $0.73, 0.70, 0.72, 0.71$ for the network's hidden layers $l = 1, 2, 3, 4$ respectively.

## 7 DISCUSSION AND CONCLUSION

We established a theoretical connection between flatness, feature robustness and, under the assumption of representative data, the generalization error. The relation between feature robustness and Hessian-based flatness measures has been established for $\kappa^l$, which takes into account the maximum eigenvalue of the Hessian, and $\kappa^l_{Tr}$, which uses the trace instead. Empirically, the measure $\kappa^l_{Tr}$ based on the trace of the Hessian shows a stronger correlation with the generalization error. This is not surprising,

since it takes into account the whole spectrum of the Hessian and every eigenvalue corresponds to a feature selection matrix of feature robustness. The tracial measure can be related to feature robustness by either bounding the maximum eigenvalue of the loss Hessian by its unnormalized trace or by averaging feature robustness over all orthogonal matrices $A \in O_m$. It is interesting to note that strong feature robustness does not exclude the possibility of adversarial examples, first observed by Szegedy et al. (2013), since large changes of loss for individual samples (i.e. adversarial examples) may be hidden in the mean in the definition of feature robustness. In Appendix C.2 we briefly discuss the freedom of perturbing individual points by suitable feature selection matrices $A$.

In contrast to existing measures of flatness, our proposed measure is invariant to layer-wise reparameterizations of ReLU networks. However, we note that other reparameterizations are possible, e.g., we can use the positive homogeneity and multiply all incoming weights into a single neuron by a positive number $\lambda > 0$ and multiply all outgoing weights of the same neuron by $1/\lambda$. While the Fisher-Rao norm suggested by Liang et al. (2019) is invariant to such reparameterizations, our proposed measures of flatness $\kappa^l$ and $\kappa^l_{Tr}$ are in general not. In principle, variations of our flatness measures can be found that are invariant to such reparameterizations as well (see Appendix B) but their analysis, except for some empirical evaluations in Appendix E, is left for future work.

The second term in the generalization bound of Theorem 10 is given by our notion of representativeness. In order to find specific bounds for the $\epsilon$-representativeness of $(S, \mathfrak{A}_\delta)$, a distribution over matrices is required that induces a distribution which is similar to a localized kernel density estimation (KDE). While our notion of representativeness is a generalization of classical representativeness, it remains open whether it is efficiently computable. The more feature robust a model is, the more freedom there is to finding specific distributions over matrices that lead to bounds on the generalization error. In Appendix D we give a computation of representativeness for a KDE with Gaussian kernels.

Taking things together, we proposed a novel and practically useful flatness measure that strongly correlates with the generalization error. We theoretically investigated this connection by relating this measure to feature robustness. This notion of robustness, together with a novel notion of representativeness provides a link to the generalization error. To the best of our knowledge, this yields the first theoretical connection between a notion of robustness, flatness of the loss surface, and generalization error and can help to better understand the performance of deep neural networks.

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

# A PROOFS OF MAIN RESULTS

## A.1 PROOF OF EQUATION 6

First note that for $||A|| \leq 1$,

$$
\begin{aligned}
||wA||_F = \left\| \begin{pmatrix} w_1 \\ w_2 \\ \vdots \\ w_m \end{pmatrix} A \right\|_F &= \left\| \begin{pmatrix} w_1 A \\ w_2 A \\ \vdots \\ w_m A \end{pmatrix} \right\|_F \\
&= \sqrt{\sum_{j=1}^m ||w_j A||_2^2} \leq \sqrt{\sum_{j=1}^m ||w_j||_2^2} \\
&= ||w||_F.
\end{aligned}
\tag{15}
$$

From (4) and (5) we get

$$
\begin{aligned}
\max_{||A|| \leq 1} \mathcal{F}(\delta, S, A) &\overset{(4),(5)}{=} \max_{||A|| \leq 1} \frac{\delta^2}{2} \langle \mathbf{w}_* A, \ H\mathcal{E}_{emp}(\mathbf{w}_*, S) \cdot (\mathbf{w}_* A) \rangle + \mathcal{O}(\delta^3) \\
&\overset{(15)}{\leq} \max_{||\mathbf{z}||_2 \leq ||w_*||_F} \frac{\delta^2}{2} \mathbf{z}^T H\mathcal{E}_{emp}(\mathbf{w}_*, S) \mathbf{z} + \mathcal{O}(\delta^3) \\
&= \max_{||\mathbf{z}||_2 = 1} \frac{\delta^2}{2} ||w_*||_F^2 \ \mathbf{z}^T H\mathcal{E}_{emp}(\mathbf{w}_*, S) \mathbf{z} + \mathcal{O}(\delta^3) \\
&= \frac{\delta^2}{2} ||\mathbf{w}_*||_F^2 \ \lambda_{max}^H(\mathbf{w}_*) + \mathcal{O}(\delta^3),
\end{aligned}
$$

where we used the identity that $\max_{||x||=1} x^T M x = \lambda_{max}^M$ for any symmetric matrix $M$.

## A.2 PROOF OF THEOREM 5

In this section, we discuss the proof to Theorem 5. Before starting with the formal proof, we discuss the idea in a simplified setting to separate the essential insight from the complicated notation in the setting of neural networks.

Let $F, \tilde{F} : \mathbb{R}^d \to \mathbb{R}$ denote twice differentiable functions such that $F(w) = \tilde{F}(\lambda w)$ for all $w$ and all $\lambda > 0$. Later, $w$ will correspond to weights of a specific layer of the neural network and the functions $F$ and $\tilde{F}$ will correspond respectively to the neural network functions before and after reparameterizations of possibly all layers of the network. We show that

$$
\frac{1}{\lambda^2} H(F(w)) = H(\tilde{F}(\lambda w)).
$$

Indeed, the second derivative of $\tilde{F}$ at $\lambda w$ with respect to coordinates $w_i, w_j$ is given by the differential quotient as

$$
\begin{aligned}
\frac{\partial^2 \tilde{F}(\lambda w)}{\partial w_i \partial w_j} &= \lim_{h \to 0} \frac{\tilde{F}(\lambda w + he_i + he_j) - \tilde{F}(\lambda w + he_i) - \tilde{F}(\lambda w + he_j) + \tilde{F}(\lambda w)}{h^2} \\
&= \lim_{h \to 0} \frac{\tilde{F}(\lambda(w + \frac{h}{\lambda} e_i + \frac{h}{\lambda} e_j)) - \tilde{F}(\lambda(w + \frac{h}{\lambda} e_i)) - \tilde{F}(\lambda(w + \frac{h}{\lambda} e_j)) + \tilde{F}(\lambda w)}{\left(\frac{h}{\lambda}\right)^2 \lambda^2} \\
&= \frac{1}{\lambda^2} \lim_{h \to 0} \frac{F(w + \frac{h}{\lambda} e_i + \frac{h}{\lambda} e_j) - F(w + \frac{h}{\lambda} e_i) - F(w + \frac{h}{\lambda} e_j) + F(w)}{\left(\frac{h}{\lambda}\right)^2} \\
&= \frac{1}{\lambda^2} \frac{\partial^2 F(w)}{\partial w_i \partial w_j}.
\end{aligned}
$$

Since this holds for all combinations of coordinates, we see that $H\tilde{F}(\lambda w) = {}^{1}\!/_{\lambda^2} HF(w)$ for the Hessians of $F$ and $\tilde{F}$, and hence

$$||\lambda w||^2 H\tilde{F}(\lambda w) = \lambda^2 ||w||^2 \frac{1}{\lambda^2} HF(w) = ||w||^2 HF(w).$$

**Formal Proof of Theorem 5**   We are given a neural network function $f(x; \mathbf{w}_1, \mathbf{w}_2, \dots, \mathbf{w}_L)$ parameterized by weights $\mathbf{w}_i$ of the $i$-th layer and positive numbers $\lambda_1, \dots, \lambda_L$ such that $f(x; \mathbf{w}_1, \mathbf{w}_2, \dots, \mathbf{w}_L) = f(x; \lambda_1 \mathbf{w}_1, \lambda_2 \mathbf{w}_2, \dots, \lambda_L \mathbf{w}_L)$ for all $\mathbf{w}_i$ and all $x$. With $\mathbf{w}$ defined by $\mathbf{w} = (\mathbf{w}_1, \mathbf{w}_2, \dots, \mathbf{w}_L)$, $\mathbf{w}_l^\lambda = \lambda_l \mathbf{w}_l$ and $\mathbf{w}^\lambda = (\mathbf{w}_1^\lambda, \mathbf{w}_2^\lambda, \dots, \mathbf{w}_L^\lambda)$, we aim to show that

$$\kappa^l(\mathbf{w}) = \kappa^l(\mathbf{w}^\lambda),$$

where $\kappa^l(\mathbf{w}) = ||\mathbf{w}_l||^2 \, \lambda_{max}^{H,l}(\mathbf{w}_l)$ is the product of the squared norm of vectorized weight matrix $\mathbf{w}_l$ with the maximal eigenvalue of the Hessian of the empirical error at $\mathbf{w}$ with respect to parameters $\mathbf{w}_l$.

Let $F(\mathbf{u}) := \sum_{(x,y) \in S} \ell(f(x; \mathbf{w}_1, \mathbf{w}_2, \dots, \mathbf{u}, \dots, \mathbf{w}_L), y)$ denote the loss as a function on the parameters of the $l$-th layer before reparameterization. Further, we let $\tilde{F}(\mathbf{v}) := \sum_{(x,y) \in S} \ell(f(x; \mathbf{w}_1^\lambda, \mathbf{w}_2^\lambda, \dots, \mathbf{v}, \dots, \mathbf{w}_L^\lambda), y)$ denote the loss as a function on the parameters of the $l$-th layer after reparameterization. We define a linear function $\eta$ by $\eta(\mathbf{u}) = \lambda_l \mathbf{u}$. By assumption, we have that $\tilde{F}(\eta(\mathbf{w}_l)) = F(\mathbf{w}_l)$ for all $\mathbf{w}_l$. By the chain rule, we compute for any variable $u^{(i,j)}$ of $\mathbf{u}$,

$$\frac{\partial F(\mathbf{u})}{\partial u^{(i,j)}}\bigg|_{\mathbf{u}=\mathbf{w}_l} = \frac{\partial \tilde{F}(\eta(\mathbf{u}))}{\partial u^{(i,j)}}\bigg|_{\mathbf{u}=\mathbf{w}_l}$$

$$= \sum_{k,m} \frac{\partial \tilde{F}(\eta(\mathbf{u}))}{\partial (\eta(\mathbf{u})^{(k,m)})}\bigg|_{\eta(\mathbf{u})=\eta(\mathbf{w}_l)} \cdot \frac{\partial (\eta(\mathbf{u})^{(k,m)})}{\partial u^{(i,j)}}\bigg|_{\eta(\mathbf{u})=\eta(\mathbf{w}_l)}$$

$$= \frac{\partial \tilde{F}(\mathbf{v})}{\partial v^{(i,j)}}\bigg|_{\mathbf{v}=\lambda_l \mathbf{w}_l} \cdot \lambda_l.$$

Similarily, for second derivatives, we get for all $i, j, s, t$,

$$\frac{\partial^2 F(\mathbf{u})}{\partial u^{(i,j)} \partial u^{(s,t)}}\bigg|_{\mathbf{u}=\mathbf{w}_l} = \lambda_l^2 \frac{\partial \tilde{F}(\mathbf{v})}{\partial v^{(i,j)} \partial v^{(s,t)}}\bigg|_{\mathbf{v}=\lambda_l \mathbf{w}_l},$$

Consequently, the Hessian $H$ of the empirical error before reparameterization and the Hessian $\tilde{H}$ after reparameterization satisfy $H(\mathbf{w}_l, S) = \lambda_l^2 \cdot \tilde{H}(\lambda_l \mathbf{w}_l, S)$ and also $\lambda_{max}^{H,l}(\mathbf{w}_l) = \lambda_l^2 \cdot \lambda_{max}^{\tilde{H},l}(\lambda_l \mathbf{w}_l)$. Therefore,

$$\kappa^l(\mathbf{w}) = ||\mathbf{w}_l||^2 \lambda_{max}^{H,l}(\mathbf{w}_l) = ||\mathbf{w}_l||^2 \lambda_l^2 \cdot \lambda_{max}^{\tilde{H},l}(\lambda_l \mathbf{w}_l) = ||\lambda_l \mathbf{w}||^2 \lambda_{max}^{\tilde{H},l}(\lambda_l \mathbf{w}_l) = \kappa^l(\mathbf{w}^\lambda).$$

### A.3   PROOF OF THEOREM 6

*Proof.* (i) This is just a corollary of Theorem 2 using the trivial bound that the maximal eigenvalue is bounded by the unnormalized trace (sum of eigenvalues) for positive semidefinte matrices (where all eigenvalues are positive).

(ii) We consider the set of orthogonal matrices $A \in O_m$ as equipped with the (unique) normalized Haar measure. (For the definition of the Haar measure, see e.g. Krantz & Parks (2008).) We need to show that $\mathbb{E}_{A \sim O_m} [\mathcal{F}(\delta, S, A)] \leq \cdot \frac{\delta^2}{2m} ||\mathbf{w}_*||_F^2 \, Tr(H\mathcal{E}_{emp}(\mathbf{w}_*)) + \mathcal{O}(\delta^3)$ with $\mathcal{F}(\delta, S, A)$ defined as in (2). Using (4) and (5) we get, similarly to (6),

$$\mathbb{E}_{A \sim O_m} [\mathcal{F}(\delta, S, A)] \leq \mathbb{E}_{A \sim O_m} \left[ \frac{\delta^2}{2} \langle \mathbf{w}_* A, \, H\mathcal{E}_{emp}(\mathbf{w}_*, S) \cdot (\mathbf{w}_* A) \rangle + \mathcal{O}(\delta^3) \right]$$

with $\langle \cdot, \cdot \rangle$ the scalar product with vectorized versions of

$$\mathbf{w}_* A = \begin{pmatrix} \mathbf{w}_{1*} \\ \vdots \\ \mathbf{w}_{d*} \end{pmatrix} A = \begin{pmatrix} \mathbf{w}_{1*} A \\ \vdots \\ \mathbf{w}_{d*} A \end{pmatrix}, \quad \mathbf{w}_{i*} \in \mathbb{R}^{1 \times m}.$$

We consider the vectorization of $\mathbf{w}_* A \in \mathbb{R}^{dm}$ given by $(\mathbf{w}_{1*}, \ldots, \mathbf{w}_{d*})^T$. By Lemma 11 below, we get

$$
\begin{aligned}
\mathbb{E}_{A \sim O_m}\left[\mathcal{F}(\delta, S, A)\right] &\leq \mathbb{E}_{A \sim O_m}\left[\frac{\delta^2}{2} \cdot \sum_{i,j=1}^{d} (\mathbf{w}_{i*} A) H \mathcal{E}_{emp}(\mathbf{w}_{j*}, S)(\mathbf{w}_{i*} A)^T + \mathcal{O}(\delta^3)\right] \\
&= \frac{\delta^2}{2} \cdot \sum_{i,j=1}^{d} \mathbb{E}_{A \sim O_m}\left[(\mathbf{w}_{i*} A) H \mathcal{E}_{emp}(\mathbf{w}_{j*}, S)(\mathbf{w}_{i*} A)^T\right] + \mathcal{O}(\delta^3)
\end{aligned}
\tag{16}
$$

Here, the notation $H\mathcal{E}_{emp}(\mathbf{w}_{j*}, S)$ refers to the empirical error at $\mathbf{w}_*$ but the derivatives are only taken over the parameters in the row $\mathbf{w}_{j*}$.

If $\mathbf{w}_{i*} \neq 0$, then by Proposition 3.2.1 of Krantz & Parks (2008) and the change of variables formula for measures, we get

$$
\mathbb{E}_{A \sim O_m}\left[(\mathbf{w}_{i*} A) H \mathcal{E}_{emp}(\mathbf{w}_{j*}, S)(\mathbf{w}_{i*} A)^T\right] = ||\mathbf{w}_{i*}||^2 \, \mathbb{E}_{z \in \mathbb{R}^m, ||z||=1}\left[z^T H \mathcal{E}_{emp}(\mathbf{w}_{j*}, S) z\right] \tag{17}
$$

for all $1 \leq i, j \leq d$, where the latter expectation is taken over the normalized (uniform) Hausdorff measure over the sphere $S^{m-1} \subset \mathbb{R}^m$. Now, using the unnormalized trace $Tr([h_{i,j}]) = \sum_i h_{i,i}$ we compute with the help of the so-called Hutchinson's trick:

$$
\begin{aligned}
\mathbb{E}_{z \in \mathbb{R}^m, ||z||=1}\left[z^T H \mathcal{E}_{emp}(\mathbf{w}_{j*}, S) z\right] &= \mathbb{E}_{||z||=1}\left[Tr(z^T H \mathcal{E}_{emp}(\mathbf{w}_{j*}, S) z)\right] \\
&= \mathbb{E}_{||z||=1}\left[Tr(H \mathcal{E}_{emp}(\mathbf{w}_{j*}, S) z z^T)\right] \\
&= Tr(H \mathcal{E}_{emp}(\mathbf{w}_{j*}, S) \, \mathbb{E}_{||z||=1}\left[z z^T\right]).
\end{aligned}
\tag{18}
$$

Note that $zz^T = [z_i z_j]_{i,j}$ and due to symmetry $\mathbb{E}_{||z||=1}[z_i z_j] = \mathbb{E}_{||z||=1}[z_i(-z_j)]$ for $i \neq j$, hence $\mathbb{E}_{||z||=1}[z_i z_j] = 0$ whenever $i \neq j$. Further $\mathbb{E}_{||z||=1}[z_i^2] = \frac{1}{m} \mathbb{E}_{||z||=1}\left[\sum_{i=1}^{m} z_i^2\right] = \frac{1}{m} \mathbb{E}_{||z||=1}\left[||z||^2\right] = \frac{1}{m}$ for all $i$. Therefore $\mathbb{E}_{||z||=1}\left[zz^T\right] = \frac{1}{m} \cdot I_m$ is a constant multiple of the identity matrix. Putting things together we have

$$
\begin{aligned}
\mathbb{E}_{A \sim O_m}\left[\mathcal{F}(\delta, S, A)\right] &\overset{(16)}{\leq} \frac{\delta^2}{2} \cdot \sum_{i,j=1}^{d} \mathbb{E}_{A \sim O_m}\left[(\mathbf{w}_{i*} A) H \mathcal{E}_{emp}(\mathbf{w}_{j*}, S)(\mathbf{w}_{i*} A)^T\right] + \mathcal{O}(\delta^3) \\
&\overset{(17)}{\leq} \frac{\delta^2}{2} \cdot \sum_{i=1,j}^{d} ||\mathbf{w}_{i*}||^2 \, \mathbb{E}_{||z||=1}\left[z^T H \mathcal{E}_{emp}(\mathbf{w}_{j*}, S) z\right] + \mathcal{O}(\delta^3) \\
&\overset{(18)}{=} \frac{\delta^2}{2} \cdot \sum_{i,j=1}^{d} ||\mathbf{w}_{i*}||^2 \frac{1}{m} \cdot Tr(H \mathcal{E}_{emp}(\mathbf{w}_{j*}, S)) + \mathcal{O}(\delta^3) \\
&= \frac{\delta^2}{2} \cdot \left(\sum_{i=1}^{d} ||\mathbf{w}_{i*}||^2\right) \cdot \left(\sum_{j=1}^{d} \frac{1}{m} \cdot Tr(H \mathcal{E}_{emp}(\mathbf{w}_{j*}, S))\right) + \mathcal{O}(\delta^3) \\
&= \frac{\delta^2}{2} \left(||\mathbf{w}_*||_F^2\right) \cdot \left(\frac{1}{m} Tr(H \mathcal{E}_{emp}(\mathbf{w}_*, S))\right) + \mathcal{O}(\delta^3) \\
&= \frac{\delta^2}{2m} ||\mathbf{w}_*||_F^2 \cdot Tr(H \mathcal{E}_{emp}(\mathbf{w}_*, S)) + \mathcal{O}(\delta^3).
\end{aligned}
$$

$\square$

**Lemma 11.** *(i) Let $H = [H_{i,j}]_{i,j}$ be a positive semidefinite matrix in $\mathbb{R}^{2m \times 2m}$ that consists of submatrices $H_{i,j} \in \mathbb{R}^{m \times m}$, $1 \leq i, j \leq 2$. Then for all $x = \begin{pmatrix} x_1 \\ x_2 \end{pmatrix} \in \mathbb{R}^{2m}$ with $x_i \in \mathbb{R}^m$, we have*
$2x_1^T H_{1,2} x_2 \leq x_1^T H_{2,2} x_1 + x_2^T H_{1,1} x_2$.
*(ii) Let $d, m \in \mathbb{N}$ and $H = [H_{i,j}]_{i,j}$ be a positive definite matrix in $\mathbb{R}^{dm \times dm}$ that consists of submatrices $H_{i,j} \in \mathbb{R}^{m \times m}$, $1 \leq i, j \leq d$. Then for all $x = (x_1, \ldots, x_d) \in \mathbb{R}^{dm}$ with $x_i \in \mathbb{R}^m$, we have $x^T H x \leq \sum_{i,j=1}^{d} x_j^T H_{i,i} x_j$.*

*Proof.* (i) By definition, $H$ is positive semidefinite if ($H$ is symmetric and) $z^T H z \geq 0$ for all $z$. Choosing $z = (-x_2, x_1)$ gives $x_2^T H_{1,1} x_2 + x_1^T H_{2,2} x_1 - 2x_1^T H1, 2x_2 \geq 0$, hence $2x_1^T H_{1,2} x_2 \leq x_1^T H_{2,2} x_1 + x_2^T H_{1,1} x_2$.

(ii) Using that every submatrix $H_{a,b} = \begin{pmatrix} H_{a,a} & H_{a,b} \\ H_{a,b}^T & H_{b,b} \end{pmatrix}$ is positive definite together with (i), we obtain

$$x^T H x = \sum_i x_i^T H_{i,i} x_i + \sum_{i \neq j} 2x_i^T H_{i,j} x_j$$

$$\leq \sum_i x_i^T H_{i,i} x_i + \sum_{i \neq j} \left( x_i^T H_{j,j} x_i + x_j^T H_{i,i} x_j \right) = \sum_{i,j} x_i^T H_{j,j} x_i$$

$\square$

### A.4 PROOF OF THEOREM 10

We are given a function $f(x) = (\psi \circ \phi)(x)$. By assumption, $f$ is $((\delta, S, A), \epsilon)$-feature robust for all matrices $||A|| \leq 1$, which implies that

$$\left| \frac{1}{|S|} \sum_{(x_i, y_i) \in S} [\ell(\psi(\phi(x_i) + \delta A \phi(x_i)), y_i) - \ell(f(x_i), y_i)] \right| \leq \epsilon \text{ for all } ||A|| \leq 1. \tag{19}$$

Further, we are given that $\phi(S)$ is $(\epsilon', \mathfrak{A}_\delta)$-representative for a hypothesis space $\mathcal{H}$ such that $\psi \in \mathcal{H}$. By Definition 9 (ii) this means that there is some $\Lambda_{\delta, \mathcal{A}} = (\lambda_i, \delta_0)_i \in \mathfrak{A}_\delta$ such that $(S, \Lambda_{\delta, \mathcal{A}})$ is $\epsilon'$-representative for $\psi$. That is, by Definition 9 (i),

$$\left| \mathbb{E}_{(x,y) \sim \mathcal{D}} [\ell(f(x), y)] - \frac{1}{|S|} \sum_{(x_i, y_i) \in S} \mathbb{E}_{\xi_x \sim \lambda_i} [\ell(\psi(\phi(x_i) + \xi_x), y_i)] \right| \leq \epsilon'. \tag{20}$$

Since $\Lambda_{\delta, \mathcal{A}} = (\lambda_i, \delta_0)_i \in \mathfrak{A}_\delta$, there exists a probability distribution $\mathcal{A}$ of matrices $||A|| \leq 1$ (so that $||\delta A|| \leq \delta$) such that

$$\frac{1}{|S|} \sum_{(x_i, y_i) \in S} \mathbb{E}_{\xi_x \sim \lambda_i} [\ell(f(\phi(x_i) + \xi_x), y_i)] = \frac{1}{|S|} \sum_{(x_i, y_i) \in S} \mathbb{E}_{A \sim \mathcal{A}} [\ell(\psi(\phi(x_i) + \delta A \phi(x_i)), y_i)]$$

$$= \mathbb{E}_{A \sim \mathcal{A}} \left[ \frac{1}{|S|} \sum_{(x_i, y_i) \in S} \ell(\psi(\phi(x_i) + \delta A \phi(x_i)), y_i) \right]. \tag{21}$$

Putting things together, we get for the generalization error $\mathcal{E}_{gen}(f, S)$ of model $f$,

$$\mathcal{E}_{gen}(f, S) = \left| \mathbb{E}_{(x,y) \sim \mathcal{D}} [\ell(f(x), y)] - \frac{1}{|S|} \sum_{(x_i, y_i) \in S} \ell(f(x_i), y_i) \right|$$

$$\overset{(21)}{\leq} \left| \mathbb{E}_{(x,y) \sim \mathcal{D}} [\ell(f(x), y)] - \frac{1}{|S|} \sum_{(x_i, y_i) \in S} \mathbb{E}_{z_i \sim \lambda_i} [\ell(\psi(\phi(x_i) + z_i), y_i)] \right|$$

$$+ \left| \mathbb{E}_{A \sim \mathcal{A}} \left[ \frac{1}{|S|} \sum_{(x_i, y_i) \in S} \ell(\psi(\phi(x_i) + \delta A \phi(x_i)), y_i) \right] - \frac{1}{|S|} \sum_{(x_i, y_i) \in S} \ell(f(x_i), y_i) \right|$$

$$= \left| \mathbb{E}_{(x,y) \sim \mathcal{D}} [\ell(f(x), y)] - \frac{1}{|S|} \sum_{(x_i, y_i) \in S} \mathbb{E}_{z_i \sim \lambda_i} [\ell(\psi(\phi(x_i) + z_i), y_i)] \right|$$

$$+ \left| \mathbb{E}_{A \sim \mathcal{A}} \left[ \frac{1}{|S|} \sum_{(x_i, y_i) \in S} [\ell(\psi(\phi(x_i) + \delta A \phi(x_i)), y_i) - \ell(f(x_i), y_i)] \right] \right|$$

$$\overset{(19),(20)}{\leq} \epsilon' + \epsilon.$$

# B ADDITIONAL MEASURES OF FLATNESS

We present additional measures of flatness we have considered during our study. The original motivation to study additional measures was given by the observation that there are other possible reparameterizations of a fully connected ReLU network than suitable multiplication of layers by positive scalars: We can use the positive homogeneity and multiply all incoming weights into a single neuron by a positive number $\lambda > 0$ and multiply all outgoing weights of the same neuron by $1/\lambda$. Our previous measures of flatness $\kappa^l$ and $\kappa^l_{Tr}$ are in general not independent of the latter reparameterizations. We therefore consider, for a layer $l$ of size $n_l$, feature robustness only for projection matrices $E_j \in \mathbb{R}^{n_l \times n_l}$ having zeros everywhere except a one at position $(j, j)$. At a local minimum $\mathbf{w}_*$ of the empirical error, this leads to

$$\mathcal{E}_{emp}(\mathbf{w}_{l*} + \delta \mathbf{w}_{l*} E_j, S) - \mathcal{E}_{emp}(\mathbf{w}_{l*}, S) = \frac{\delta^2}{2} \mathbf{w}_{l*}(j)^T H \mathcal{E}_{emp}(\mathbf{w}_{l*}(j), S) \mathbf{w}_{l*}(j) + \mathcal{O}(\delta^3)$$

where $\mathbf{w}_{l*}(j)$ denotes the $j$-th column vector of weight matrix $\mathbf{w}_l$ of layer $l$, and we only consider the Hessian with respect to these weight parameters. We define for each layer $l$ and neuron $j$ in that layer a flatness measure by

$$\rho^l(j)(\mathbf{w}_*) := \mathbf{w}_{l*}(j)^T H \mathcal{E}_{emp}(\mathbf{w}_{l*}(j)) \mathbf{w}_{l*}(j)$$

For each $l$ and $j$, this measure is invariant under all linear reparameterizations that do not change the network function. The proof of the following theorem is given in Section B.1

**Theorem 12.** *Let $f = f(\mathbf{w}_1, \mathbf{w}_2, \ldots, \mathbf{w}_L)$ denote a neural network function parameterized by weights $\mathbf{w}_i$ of the $i$-th layer. Suppose there are positive numbers $\lambda_1^{(i,j)}, \ldots, \lambda_L^{(i,j)}$ such that the products $\mathbf{w}_l^\lambda$ obtained from multiplying weight $w_l^{(i,j)}$ at matrix position $(i, j)$ in layer $l$ by $\lambda_l^{(i,j)}$ satisfy that $f(\mathbf{w}_1, \mathbf{w}_2, \ldots, \mathbf{w}_L) = f(\mathbf{w}_1^\lambda, \mathbf{w}_2^\lambda, \ldots, \mathbf{w}_L^\lambda)$ for all $\mathbf{w}_i$. Then $\rho^l(j)(\mathbf{w}) = \rho^l(j)(\mathbf{w}^\lambda)$ for each $j$ and $l$.*

We define a measure of flatness for a full layer by combinations of the measures of flatness for each individual neuron.

$$\rho^l(\mathbf{w}_*) := \max_j \rho^l(j)(\mathbf{w}_*) \text{ and } \rho^l_\sigma(\mathbf{w}_*) := \sum_j \rho^l(j)(\mathbf{w}_*)$$

Since each of the individual expressions is invariant under all linear reparameterizations, so are the maximum and sum.

Analogous to Theorem 2, we get an upper bound for feature robustness for projection matrices $E_j$.

**Theorem 13.** *Let $f$ denote a neural network function of a $L$-layer fully connected neural network. For each layer $l, 1 \leq l \leq L$ of size $n_l$ let $E_j \in \mathbb{R}^{n_l \times n_l}$ denote the projection matrix containing only zeros except a 1 at position $(j, j)$. Let $\mathbf{w}_{l*}$ denote weights of the $l$-th layer at a local minimum of the empirical error.*

*Then the neural network is $\left((\delta, S, E_j), \delta^2/2 \rho^l(\mathbf{w}_*) + \mathcal{O}(\delta^3)\right)$-feature robust for all $j$ at $\mathbf{w}_*$.*

**One Value for all layers** Our measure of flatness are strongly related to feature robustness, which evaluates the sensitivity toward small changes of features. In a good predictor, generalization behavior should correlate with the amount of change of the loss under changes of discriminating features. For neural networks, we can consider the output of each layer as a feature representation. Each flatness measure $\kappa^l$ is then related by Corollary 13 to changes of the features of the $l$-th layer. It is however clear that a low value of $\kappa^l$ for a specific layer $l$ alone cannot explain good performance. We therefore specify a common bound for all layers.

Denoting by $\mathbf{w}_*$ the set of weights from all layers combined, we have $||\mathbf{w}_*^l||_F \leq ||\mathbf{w}_*||_F$ for all $l$. Further, if $H(l)$ denotes the Hessian of the loss with respect to only the weights of the $l$-th layer, and $H$ the Hessian with respect to the weights of all layers, then $\lambda_{max}^{H(l),l}(\mathbf{w}_*^l) \leq \lambda_{max}^H(\mathbf{w}_*)$. (This holds since

$$\lambda(A) = \max_{||v||=1} v^T A v \text{ and } (v, 0)^T \begin{pmatrix} A & D \\ D^T & B \end{pmatrix} \begin{pmatrix} v \\ 0 \end{pmatrix} = v^T A v.)$$

Table 1: Hessian based measures of flatness

| Notation | Definition | One value per | Invariance |
|---|---|---|---|
| $\kappa$ | $\|\vec{\mathbf{w}}\|^2 \cdot \lambda_{max}^H(\vec{\mathbf{w}})$ | network | none |
| $\kappa^l$ | $\|\mathbf{w}_l\|^2 \cdot \lambda_{max}^{H,l}(\mathbf{w}_l)$ | layer | layer-wise mult. by pos scalar |
| $\kappa_{Tr}^l$ | $\|\mathbf{w}\|^2 \cdot Tr(H\mathcal{E}_{emp}(\mathbf{w}_l, S))$ | layer | layer-wise mult. by pos scalar |
| $\kappa^{max}$ | $\max_l \kappa^l(\mathbf{w})$ | network | layer-wise mult. by pos scalar |
| $\kappa^\Sigma$ | $\sum_{l=1}^L \kappa^l(\mathbf{w})$ | network | layer-wise mult. by pos scalar |
| $\kappa_{Tr}^{max}$ | $\max_l \kappa_{Tr}^l(\mathbf{w})$ | network | layer-wise mult. by pos scalar |
| $\kappa_{Tr}^\Sigma$ | $\sum_{l=1}^L \kappa_{Tr}^l(\mathbf{w})$ | network | layer-wise mult. by pos scalar |
| $\rho^l(j)$ | $\mathbf{w}_l(j)^T H\mathcal{E}_{emp}(\mathbf{w}_l(j), S)\mathbf{w}_l(j)$ | neuron | all linear reparameterizations |
| $\rho^l$ | $\max_j \rho^l(j)(\mathbf{w})$ | layer | all linear reparameterizations |
| $\rho_\sigma^l$ | $\sum_j \rho^l(j)(\mathbf{w})$ | layer | all linear reparameterizations |
| $\rho^{max}$ | $\max_l \rho^l(\mathbf{w})$ | network | all linear reparameterizations |
| $\rho^\Sigma$ | $\sum_{l=1}^L \rho_\sigma^l(\mathbf{w})$ | network | all linear reparameterizations |

Therefore, no matter which layer with activation values $\phi^l(x_i)$ for each $x_i \in S$ we are perturbing with a matrix $\|A_l\| \leq 1$ to $\phi^l(x_i) + \delta A_l \cdot \phi^l(x_i)$, we have that

$$\mathcal{F}(\delta, S, A) \leq \frac{\delta^2}{2}\|\mathbf{w}_*\|_F^2 \cdot \lambda_{max}^H(\mathbf{w}_*) + \mathcal{O}(\delta^3),$$

and $\kappa(\mathbf{w}_*) = \|\mathbf{w}_*\|_F^2 \cdot \lambda_{max}^H(\mathbf{w}_*)$ can be considered as a common measure for all layers.

However, $\kappa(\mathbf{w}_*)$ is not invariant under the reparameterizations considered in Theorem 5. We therefore consider more simple common bounds by combinations of the individual terms $\kappa^l$, e.g. by taking the maximum of $\kappa_l$ over all layers, $\kappa^{max}(\mathbf{w}_*) := \max_l \kappa^l(\mathbf{w}_*)$, or the sum $\kappa^\Sigma(\mathbf{w}_*) := \sum_{l=1}^L \kappa^l(\mathbf{w}_*)$. Since each of the individual expressions are invariant under linear reparameterizations of full layers, so are the maximum and sum.

Finally, we define $\rho^{max}(\mathbf{w}_*) := \max_l \rho^l(\mathbf{w}_*)$ and $\rho^\Sigma(\mathbf{w}_*) := \sum_{l=1}^L \rho_\sigma^l(\mathbf{w}_*)$.

Table 1 summarizes all our measures of flatness, specifying whether each measure is defined per network, layer or neuron, and whether it is invariant layer-wise multiplication by a positive scalar (as considered in Theorem 5) or invariant under all linear reparameterization (as considered in Theorem 12).

## B.1 PROOF OF THEOREM 12

As in Subsection A.2, we first present the idea in a simplified setting.

For the proof of Theorem 12 we need to consider the case when we multiply coordinates by different scalars. Let $F : \mathbb{R}^2 \to \mathbb{R}$ denote twice differentiable functions such that $F(v, w) = \tilde{F}(\lambda v, \mu w)$ for all $v \in \mathbb{R}$, $w \in \mathbb{R}$ and all $\lambda, \mu > 0$. In the formal proof, $v, w$ will correspond to two outgoing weights for a specific neuron, while again $F$ and $\tilde{F}$ correspond to network functions before and after reparameterizations of all possibly all weights of the neural network. Then

$$(v, w) \cdot HF(v, w) \cdot \begin{pmatrix} v \\ w \end{pmatrix} = (\lambda v, \mu w) \cdot HF(\lambda v, \mu w) \cdot \begin{pmatrix} \lambda v \\ \mu w \end{pmatrix}$$

for all $v, w$ and all $\lambda, \mu > 0$.

Indeed, the second derivative of $\tilde{F}$ at $(\lambda v, \mu w)$ with respect to coordinates $v, w$ is given by the differential quotient as

$$\frac{\partial^2 \tilde{F}(\lambda v, \mu w)}{\partial v \partial w} = \lim_{h,k \to 0} \frac{\tilde{F}(\lambda v + h, \mu w + ke) - \tilde{F}(\lambda v + h, \mu w) - \tilde{F}(\lambda v, \mu w + k) + \tilde{F}(\lambda v, w)}{hk}$$

$$= \lim_{h,k \to 0} \frac{\tilde{F}(\lambda(v + \frac{h}{\lambda}), \mu(w + \frac{k}{\mu})) - \tilde{F}(\lambda(v + \frac{h}{\lambda}, \mu w)) - \tilde{F}(\lambda v, \mu(w + \frac{k}{\lambda})) + \tilde{F}(\lambda v, \mu w)}{\left(\frac{h}{\lambda}\right)\left(\frac{k}{\mu}\right) \lambda \mu}$$

$$= \frac{1}{\lambda \mu} \lim_{h,k \to 0} \frac{F(v + \frac{h}{\lambda}, w + \frac{k}{\mu}) - F(v + \frac{h}{\lambda}, w) - F(v, w + \frac{k}{\mu}) + F(v, w)}{\frac{h}{\lambda}\frac{k}{\mu}}$$

$$= \frac{1}{\lambda \mu} \frac{\partial^2 F(v, w)}{\partial v \partial w}.$$

From the calculation above, we also see that

$$\frac{\partial^2 \tilde{F}(\lambda v, \mu w)}{\partial v \partial v} = \frac{1}{\lambda^2} \frac{\partial^2 F(v, w)}{\partial v \partial v}, \text{ and } \frac{\partial^2 \tilde{F}(\lambda v, \mu w)}{\partial w \partial w} = \frac{1}{\mu^2} \frac{\partial^2 F(v, w)}{\partial w \partial w}.$$

It follows that

$$(v, w) \cdot HF(v, w) \cdot \begin{pmatrix} v \\ w \end{pmatrix} = v^2 \frac{\partial^2 F(v, w)}{\partial v \partial v} + 2vw \frac{\partial^2 F(v, w)}{\partial v \partial w} + w^2 \frac{\partial^2 F(v, w)}{\partial w \partial w}$$

$$= (\lambda v)^2 \frac{\partial^2 \tilde{F}(v, w)}{\partial v \partial v} + 2(\lambda v)(\mu w) \frac{\partial^2 \tilde{F}(v, w)}{\partial v \partial w} + (\mu w)^2 \frac{\partial^2 \tilde{F}(v, w)}{\partial w \partial w}$$

$$= (\lambda v, \mu w) \cdot HF(\lambda v, \mu w) \cdot \begin{pmatrix} \lambda v \\ \mu w \end{pmatrix}.$$

**Formal Proof of Theorem 12**   We are given a neural network function $f(x; \mathbf{w}_1, \mathbf{w}_2, \ldots, \mathbf{w}_L)$ parameterized by weights $\mathbf{w}_i$ of the $i$-th layer and positive numbers $\lambda_1^{(i,j)}, \ldots, \lambda_L^{(i,j)}$ such that the products $\mathbf{w}_l^\lambda$ obtained from multiplying weight $w_l^{(i,j)}$ at matrix position $(i, j)$ in layer $l$ by $\lambda_l^{(i,j)}$ satisfies that $f(x; \mathbf{w}_1, \mathbf{w}_2, \ldots, \mathbf{w}_L) = f(x; \mathbf{w}_1^\lambda, \mathbf{w}_2^\lambda, \ldots, \mathbf{w}_L^\lambda)$ for all $\mathbf{w}_i$ and all $x$. We aim to show that

$$\rho^l(j)(\mathbf{w}) = \rho^l(j)(\mathbf{w}^\lambda)$$

for each $j$ and $l$ where $\rho^l(j)(\mathbf{w}) = \mathbf{w}_l(j)^T H\mathcal{E}_{emp}(\mathbf{w}_l(j), S)\mathbf{w}_l(j)$, $\mathbf{w}_l(j)$ denotes the $j$-th column of the weight matrix at the $l$-th layer and $H\mathcal{E}_{emp}(\mathbf{w}_l(j), S)$ denotes the Hessian of the empirical error with respect to the weight parameters in $\mathbf{w}_l(j)$. Similar to the above, we denote by $\mathbf{w}_l(j)^\lambda$ the product obtained from multiplying weight $\mathbf{w}_l(j)_i = w_l^{(i,j)}$ at matrix position $(i, j)$ in layer $l$ by $\lambda^{(i,j)}$.

The proof is very similar to the proof of Theorem 5, only this time we have to take the different parameters $\lambda_l^{(i,j)}$ into account. For fixed layer $l$, we denote the $j$-th column of $\mathbf{w}_l$ and $\mathbf{w}_l(j)$.

Let

$$F(\mathbf{u}) := \sum_{(x,y) \in S} \ell(f(x; \mathbf{w}_1, \mathbf{w}_2, \ldots, [\mathbf{w}_l(1), \ldots, \mathbf{w}_l(j-1), \mathbf{u}, \mathbf{w}_l(j+1), \ldots \mathbf{w}_l(n_l)],$$

$$\ldots, \mathbf{w}_L), y)$$

denote the loss as a function on the parameters of the $j$-th column in the $l$-th layer before reparameterization and

$$\tilde{F}(\mathbf{v}) := \sum_{(x,y) \in S} \ell(f(x_i; \mathbf{w}_1^{\lambda_1}, \mathbf{w}_2^{\lambda_2}, \ldots, [\mathbf{w}_l(1)^\lambda, \ldots, \mathbf{w}_l(j-1)^\lambda, \mathbf{v}, \mathbf{w}_l(j+1)^\lambda, \ldots w_l(n_l)^\lambda],$$

$$\ldots, \mathbf{w}_L{}^{\lambda_L}), y)$$

denote the loss as a function on the parameters of the $j$-th neuron in the $l$-th layer after reparameterization.

We define a linear function $\eta$ by

$$\eta(\mathbf{u}) = \eta(u_1, u_2, \ldots u_{n_l}) = \eta(u_1 \lambda_l^{(1,j)}, u_2 \lambda_l^{(2,j)}, \ldots u_{n_l} \lambda_l^{(n,j)}).$$

By assumption, we have that $\tilde{F}(\eta(\mathbf{w}_l(j))) = F(\mathbf{w}_l(j))$ for all $\mathbf{w}_l(j)$. By the chain rule, we compute for any variable $u_i$ of $\mathbf{u}$,

$$\frac{\partial F(\mathbf{u})}{\partial u_i}\Big|_{\mathbf{u}=\mathbf{w}_l(j)} = \frac{\partial \tilde{F}(\eta(\mathbf{u}))}{\partial u_i}\Big|_{\mathbf{u}=\mathbf{w}_l(j)}$$

$$= \sum_k \frac{\partial \tilde{F}(\eta(\mathbf{u}))}{\partial(\eta(\mathbf{u})_k)}\Big|_{\eta(\mathbf{u})=\eta(\mathbf{w}_l(j))} \cdot \frac{\partial(\eta(\mathbf{u})_k)}{\partial u_i}\Big|_{\eta(\mathbf{u})=\eta(\mathbf{w}_l(j))}$$

$$= \frac{\partial \tilde{F}(\mathbf{v})}{\partial v_i}\Big|_{\mathbf{v}=\mathbf{w}_l(j)^\lambda} \cdot \lambda_l^{(i,j)}.$$

Similarly, for second derivatives, we get for all $i, s$,

$$\frac{\partial^2 F(\mathbf{u})}{\partial u_i \partial u_s}\Big|_{\mathbf{u}=\mathbf{w}_l(j)} = \lambda_l^{(i,j)} \lambda_l^{(s,j)} \frac{\partial \tilde{F}(\mathbf{v})}{\partial v_i \partial v_j}\Big|_{\mathbf{v}=\mathbf{w}_l(j)^\lambda}.$$

Consequently, the Hessian $HF$ of the empirical error before reparameterization and the Hessian $H\tilde{F}$ after reparameterization satisfy that at position $(i, s)$ of the Hessian matrix,

$$HF(\mathbf{w}_l)_{(i,s)} = \lambda_l^{(i,j)} \lambda_l^{(s,j)} \cdot H\tilde{F}(\mathbf{w}_l^\lambda)_{(i,s)}.$$

Therefore,

$$\rho^l(j)(\mathbf{w}) = \mathbf{w}_l(j)^T \cdot HF(\mathbf{w}_l) \cdot \mathbf{w}_l(j) = \sum_{i,s} w_l^{(i,j)} w_l^{(s,j)} HF(\mathbf{w}_l)_{(i,s)}$$

$$= \sum_{i,s} w_l^{(i,j)} w_l^{(s,j)} \lambda_l^{(i,j)} \lambda_l^{(s,j)} \cdot H\tilde{F}(\mathbf{w}_l^\lambda)_{(i,s)}$$

$$= \sum_{i,s} \lambda_l^{((i,j)} w_l^{i,j)} \lambda_l^{(s,j)} w_l^{(s,j)} \cdot H\tilde{F}(\mathbf{w}_l^\lambda)_{(i,s)}$$

$$= (\mathbf{w}_l(j)^\lambda)^T \cdot H\tilde{F}(\mathbf{w}_l^\lambda) \cdot \mathbf{w}_l(j)^\lambda = \rho^l(j)(\mathbf{w}^\lambda)$$

## C  ADDITIONAL PROPERTIES OF FEATURE ROBUSTNESS

### C.1  RELATION TO NOISE INJECTION AT THE FEATURE SPACE

Feature robustness is related to noise injection in the layer of consideration. By defining a probability measure $\mathcal{P}_A$ on matrices $A \in \mathbb{R}^{m \times m}$ of norm $||A|| \leq 1$, we can take expectations over matrices. An expectation over such matrices induces for each sample $x \in \mathcal{X}$ an expectation over a probability distribution of vectors $\xi \in \mathbb{R}^m$ with $||\xi|| \leq ||\phi(x)||$. We find the induced probability distribution $\mathcal{P}_x$ from the measure $P_x$ defined by $P_x(T) = \mathcal{P}_A(\{A \mid A\phi(x) \in T\})$ for a measurable subset $T \subseteq \mathbb{R}^m$. Then,

$$\mathbb{E}_{A \sim \mathcal{P}_A}[\mathcal{F}(\delta, S, A)] = \mathbb{E}_{A \sim \mathcal{P}_A}\left[ \frac{1}{|S|} \sum_{(x,y) \in S} [\ell(\psi(\phi(x) + \delta A\phi(x), y)) - \ell(f(x), y)] \right]$$

$$= \frac{1}{|S|} \sum_{(x,y) \in S} \mathbb{E}_{\xi_x \in \mathcal{P}_x}[\ell(\psi(\phi(x) + \delta\xi_x) - \ell(f(x), y)].$$

The latter is robustness to noise injection according to noise distribution $\mathcal{P}_x$ for sample $x$ in the feature space defined by $\phi$.

## C.2 ADVERSARIAL EXAMPLES

**Large changes of loss (adversarial examples) can be hidden in the mean in the definition of feature robustness.** We have seen that flatness of the loss curve with respect to some weights is related to the mean change in loss value when perturbing all data points $x_i$ into directions $Ax_i$ for some matrix $A$. For a common bound over different directions governed by the matrix $A$, we restrict ourselves to matrices $||A|| \leq 1$. One may therefore wonder, what freedom of perturbing individual points do we have?

At first, note that for each fixed sample $x_{i_0}$ and direction $z_{i_0}$ there is a matrix $A$ such that $Ax_{i_0} = z_{i_0}$, so each direction for each datapoint can be considered within a bound as above. We get little insight over the change of loss for this perturbation however, since a larger change of the loss may go missing in the mean change of loss over all data points considered in the same bound.

The bound involving $\kappa(\mathbf{w}_*)$ from above does not directly allow to check the change of the loss when perturbing the samples $x_i$ independently into arbitrary directions . For example, suppose we have two samples close to each other and we are interested in the change of loss when perturbing them into directions orthogonal to each other. Specifically, suppose our dataset contains the points $(1, 0, 0, \ldots, 0)$ and $(1, \epsilon, 0, \ldots, 0)$ for some small $\epsilon$, and we aim to check how the loss changes when perturbing $(1, 0, 0, \ldots, 0)$ into direction $(1, 0, 0, \ldots, 0)$ and $(1, \epsilon, 0, \ldots, 0)$ orthogonally into direction $(0, 1, 0, \ldots, 0)$. To allow for this simultaneous change, our matrix $A$ has to be of the form

$$
A = \begin{pmatrix}
1 & -\frac{1}{\epsilon} & \cdots \\
0 & \frac{1}{\epsilon} & \cdots \\
0 & & \\
\vdots & \vdots & \\
0 & 0 & \cdots
\end{pmatrix}.
$$

Then

$$
||A|| \geq ||A \cdot \begin{pmatrix} 0 \\ 1 \\ 0 \\ \vdots, \\ 0 \end{pmatrix}|| = ||(-\frac{1}{\epsilon}, \frac{1}{\epsilon}, 0, \ldots)|| = \frac{\sqrt{2}}{\epsilon}.
$$

Hence, our desired alterations of the input necessarily lead to a large matrix norm $||A||$ and our attainable bound with $||A||^2 \kappa(\mathbf{w}_*)$ becomes almost vacuous.

## C.3 CONVOLUTIONAL LAYERS

Feature robustness is not restricted to fully connected neural networks. In this section, we briefly consider convolutional layers $\mathbf{w} * x$. Using linearity, we get $\mathbf{w} * (x + \delta x) = (\mathbf{w} + \delta \mathbf{w}) * x$. What about changes $(\mathbf{w} + \delta \mathbf{w} A)$ for some matrix $A$? Since convolution is a linear function, there is a matrix $W$ such that $\overrightarrow{\mathbf{w} * x} = Wx$ and there is a matrix $W_A$ such that $\overrightarrow{\mathbf{w} A * x} = W_A x$. We assume that the convolutional layer is dimensionality-reducing, $W \in \mathbb{R}^{n \times m}, m < n$ and that the matrix $W$ has full rank, so that there is a matrix $V$ with $WV = I_m$.[1] Then

$$
(\mathbf{w} + \delta \mathbf{w} A) * x = Wx + \delta W_A x = Wx + \delta W V W_B x = W(x + \delta V W_B x).
$$

As a consequence, similar considerations of flatness and feature robustness can be considered for convolutional layers.

---

[1]This holds for example for a convolutional filter with stride one without padding, as in this case $W$ has a Toeplitz submatrix of size $(m \times m)$.

# D  A LINK BETWEEN $\epsilon$-REPRESENTATIVENESS AND KERNEL DENSITY ESTIMATION

Compared to classical representativeness, the definition of $\epsilon$-representativeness is far more general, allowing the choice of a family of distributions $\Lambda = (\lambda_i, \nu_i)_{1 \le i \le N}$. A suitable restriction is to consider only local distributions $\lambda_i$ and $\nu_i$ centered around the origin 0. With this, the following connection to kernel density estimation can be established: If a distribution can be approximated with error $\epsilon$ by a kernel density estimation then this sample is representative.

**Proposition 14.** *(i) If a distribution $\mathcal{D}$ can be $\epsilon$-approximated by a Kernel Density Estimator with Gaussian kernels and a diagonal bandwidth matrix using a sample $S$ of size $N \in \mathbb{N}$ and if the loss $\ell : \mathcal{Y} \times \mathcal{Y} \to \mathbb{R}_+$ is bounded by $L$, then for any such sample $S$ there is some $\Lambda_N$ such that $(S, \Lambda_N)$ is $L\epsilon$-representative for $f$ with respect to $\mathcal{D}$ and $\ell$. (ii) If the probability density function of $\mathcal{D}$ lies inside a $d$-dimensional kernel Hilbert space with Gaussian kernel $K_h$, i.e., $P_{\mathcal{D}}(x, y) \in \mathcal{H}_d$, then $(S, \Lambda_N)$ is $L\epsilon$-representative with $\epsilon \in \mathcal{O}\left(N^{-1/4}\right)$.*

Before we proof this proposition it is important to note that this result—in its current form—cannot be used to obtain a generalization bound using Theorem 10: In Proposition 14, $\Lambda = (\lambda_i \times \nu_i)$ is chosen such that $P_{(\lambda_i \times \nu_i)}(z) = K_h(z)$, where $K_h$ denotes the Gaussian kernel. Theorem 10 requires the distribution to be induced by a probability distribution $\mathcal{A}$ on feature matrices $A$ with $\|A\| \le \delta$. However, since Gaussians have support everywhere, the assumption that $\|A\| \le \delta$ for any finite $\delta > 0$ does not hold. A possible solution would be to use truncated Gaussian kernels, for which $\|A\| \le \delta$ can be ensured. However, it remains an open question whether there exists a probability distribution $\mathcal{A}$ over feature matrices $A$ that induces suitable truncated Gaussian distributions which would allow to compute practical bounds on the generalization error.

We now provide the proof to Proposition 14.

*Proof.* Given a sample $S \sim \mathcal{D}$ with $|S| = N$, its representativeness is defined as

$$Rep(S) = \left| \mathbb{E}_{(x,y)\sim\mathcal{D}} \left[ \ell(f(x), y) \right] - \frac{1}{N} \sum_{(x_i, y_i) \in S} \mathbb{E}_{(\xi_x, \xi_y)\sim(\lambda_i, \nu_i)} \left[ \ell(f(x_i + \xi_x), y_i + \xi_y) \right] \right| .$$

We can rewrite $\mathbb{E}_{(x,y)\sim\mathcal{D}} \left[ \ell(f(x), y) \right]$ as

$$\mathbb{E}_{(x,y)\sim\mathcal{D}} \left[ \ell(f(x), y) \right] = \int_{(x,y)\in\mathcal{X}\times\mathcal{Y}} \ell(f(x), y) P_{\mathcal{D}}(x, y) d(x, y) = \int_{z\in\mathcal{Z}} \ell(z) P_{\mathcal{D}}(z) dz ,$$

where we abbreviate $\mathcal{X} \times \mathcal{Y} = \mathcal{Z}$, $z = (x, y)$, and with slight abuse of notation write $\ell(f(x), y) = \ell_f(x, y) = \ell(z)$ for fixed $f$. Furthermore, since $\lambda_i$ and $\nu_i$ are independent we can rewrite

$$\frac{1}{N} \sum_{(x_i, y_i)\sim S} \mathbb{E}_{(\xi_x, \xi_y)\sim(\lambda_i, \nu_i)} \left[ \ell(f(x_i + \xi_x), y_i + \xi_y) \right]$$

$$= \frac{1}{N} \sum_{(x_i, y_i)\in S} \int_{(\xi_x, \xi_y)\in\mathcal{X}\times\mathcal{Y}} \ell(f(x_i + \xi_x), y_i + \xi_y) P_{(\lambda_i \times \nu_i)}(\xi_x, \xi_y) d(\xi_x, \xi_y) .$$

$$= \frac{1}{N} \sum_{z_i \in S} \int_{\xi\in\mathcal{Z}} \ell(z_i + \xi)) P_{(\lambda_i \times \nu_i)}(\xi) d\xi$$

By assumption, a Kernel Density Estimator on sample $S$, i.e.,

$$\widehat{P}(z) = \frac{1}{N} \sum_{z_i \in S} K_h(z - z_i)$$

with kernel $K_h$, approximates $P_{\mathcal{D}}(z)$ with approximation error $\epsilon$. Thus, we get that

$$\left| \int_{z \in \mathcal{Z}} \ell(z) P_{\mathcal{D}}(z) dz - \frac{1}{N} \sum_{z_i \in S} \int_{\xi \in \mathcal{Z}} \ell(z_i + \xi) P_{(\lambda_i \times \nu_i)}(\xi) d\xi \right|$$

$$\leq \left| \int_{z \in \mathcal{Z}} \ell(z) \widehat{P}(z) dz - \frac{1}{N} \sum_{z_i \in S} \int_{\xi \in \mathcal{Z}} \ell(z_i + \xi) P_{(\lambda_i \times \nu_i)}(\xi) d\xi \right| + \epsilon \max_{z \in \mathcal{Z}} (\ell(z))$$

$$\leq \left| \int_{z \in \mathcal{Z}} \ell(z) \frac{1}{N} \sum_{z_i \in S} K_h(z - z_i) dz - \frac{1}{N} \sum_{z_i \in S} \int_{\xi \in \mathcal{Z}} \ell(z_i + \xi) P_{(\lambda_i \times \nu_i)}(\xi) d\xi \right| + \epsilon \max_{z \in \mathcal{Z}} (\ell(z))$$

$$\leq \frac{1}{N} \sum_{z_i \in S} \left| \int_{z \in \mathcal{Z}} \ell(z) K_h(z - z_i) dz - \int_{\xi \in \mathcal{Z}} \ell(z_i + \xi) P_{(\lambda_i \times \nu_i)}(\xi) d\xi \right| + \epsilon L \ .$$

By substituting $\zeta = z - z_i$ and choosing the $(\lambda_i \times \nu_i)$ such that $P_{(\lambda_i \times \nu_i)}(z) = K_h(z)$ (which is possible since we assumed the bandwidth matrix to be diagonal), we can further rewrite this as

$$Rep(S) \leq \frac{1}{N} \sum_{z_i \in S} \left| \int_{z \in \mathcal{Z}} \ell(z) K_h(z - z_i) dz - \int_{\xi \in \mathcal{Z}} \ell(z_i + \xi) P_{(\lambda_i \times \nu_i)}(\xi) d\xi \right| + \epsilon L$$

$$= \frac{1}{N} \sum_{z_i \in S} \left| \int_{\zeta \in \mathcal{Z}} \ell(\zeta + z_i) K_h(\zeta) d\zeta - \int_{\xi \in \mathcal{Z}} \ell(z_i + \xi) P_{(\lambda_i \times \nu_i)}(\xi) d\xi \right| + \epsilon L$$

$$\leq \frac{1}{N} \sum_{z_i \in S} \left| \int_{\zeta \in \mathcal{Z}} \ell(\zeta + z_i) \underbrace{\left( K_h(\zeta) - P_{(\lambda_i, \nu_i)}(\zeta) \right)}_{=0} d\zeta \right| + \epsilon L = \epsilon L$$

If the probability density function of $\mathcal{D}$ lies inside a $d$-dimensional kernel Hilbert space with Gaussian kernel $K_h$, i.e., $P_{\mathcal{D}}(x, y) \in \mathcal{H}_d$, then it follows from Theorem 4 in Fasshauer et al. (2012) that

$$\epsilon \leq \frac{\sqrt{2}}{n^{-\frac{1}{4}}} \left( 1 + \frac{1}{2n^{-\frac{1}{2}}} \right)^{\frac{1}{2}} \in \mathcal{O}\left( n^{-\frac{1}{4}} \right)$$

$\square$

## E  ADDITIONAL EXPERIMENTS

In addition to the evaluation on the CIFAR10 dataset with LeNet5 network, we also conducted experiments on the MNIST dataset. For learning with this data, we employed a custom fully connected network with ReLU activations containing 4 hidden layers with 50, 50, 50, and 30 neurons correspondingly. The output layer has 10 neurons with softmax activation. The networks were trained till convergence on the training dataset of MNIST, moreover, the configurations that achieved larger than 0.07 training error were filtered out. All the networks were initialized according to Xavier normal scheme with random seed. For obtaining different convergence minima the batch size was varied between 1000, 2000, 4000, 8000 with learning rate changed from 0.02 to 1.6 correspondingly to keep the ratio constant. All the configurations were trained with SGD. Figure 5 shows the correlation between the layer-wise flatness measure based on the trace of the Hessian for the corresponding layer. The values for all four hidden layers are calculated (the trace is not normalized) and aligned with values of generalization error (difference between normalized test error and train error). The observed correlation is strong (with $\rho \geq 0.7$) and varies slightly for different layers, nevertheless it is hard to identify the most influential layer for identifying generalization properties.

We also calculated neuron-wise flatness measures described in Sec. B for this network configurations. In Figure 6 we depicted correlation between $\rho_\sigma^l$ and generalization loss for each of the layers, and in Figure 7–between $\rho^l$ and generalization loss. The observed correlation is again significant, but compared to the previous measure we can see that it might differ considerably depending on the layer.

The network-wise flatness measures can based both on layer-wise and neuron-wise measures as defined in Sec. B. We computed $\kappa_\tau^{max}$, $\kappa_\tau^\Sigma$, $\rho^{max}$, and $\rho^\Sigma$ and depicted them in Figure 8. Interesting to note, that each of the network-wise measures has a larger correlation with generalization loss than the original neuron-wise and layer-wise measures.

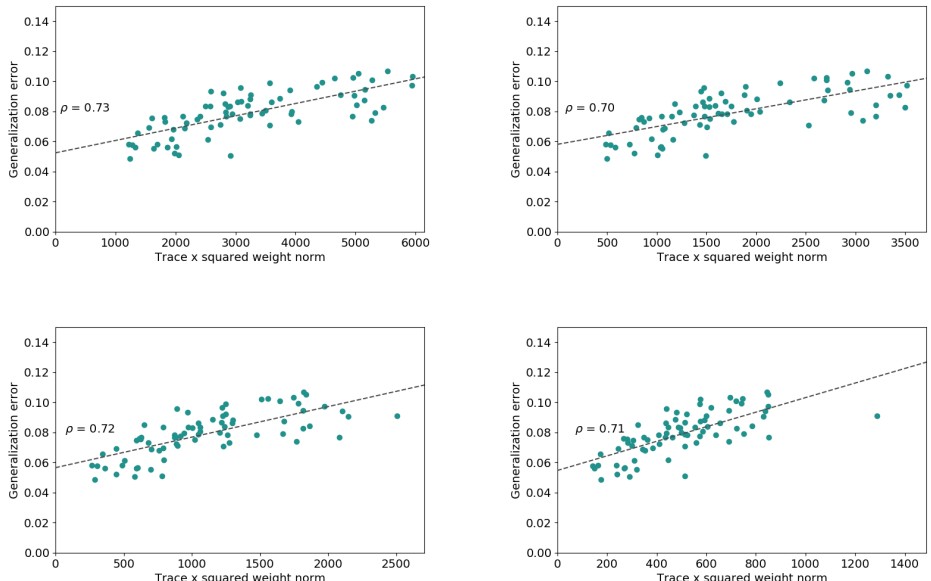

Figure 5: Layer-wise flatness measure calculated for MNIST trained fully-connected network. Four plots correspond to four hidden layers of the network. For each of the layers a strong correlation with generalization error can be observed.

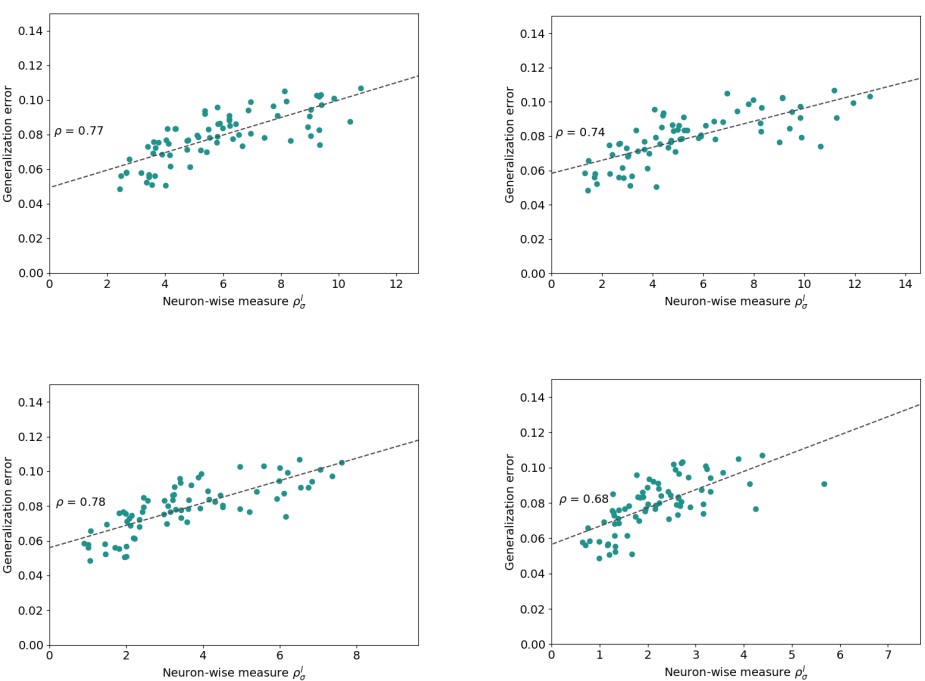

Figure 6: Neuron-wise flatness measure $\rho_\sigma^l$ calculated for each of the hidden layers for the fully-connected network trained on MNIST dataset. Each plot corresponds to a layer.

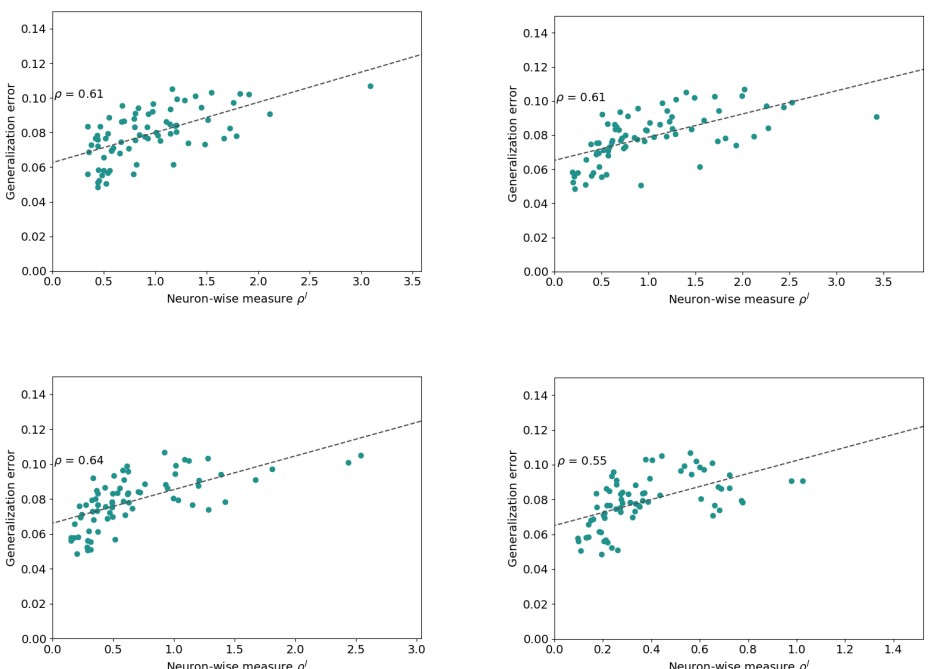

Figure 7: Neuron-wise flatness measure $\rho^l$ calculated for each of the hidden layers for the fully-connected network trained on MNIST dataset. Each plot corresponds to a layer.

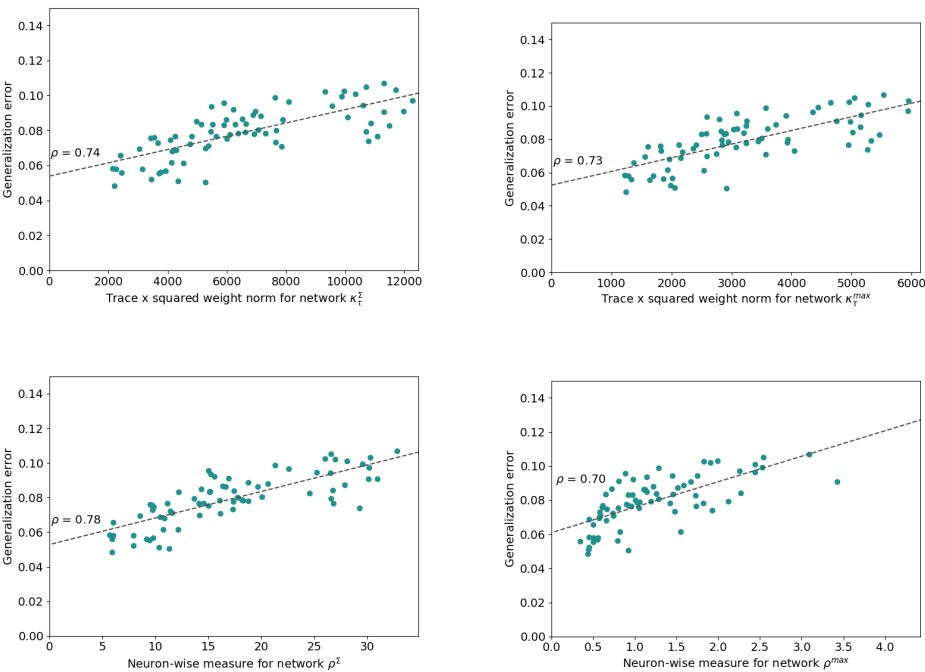

Figure 8: Network-wise flatness measures based on various neuron-wise and trace layer-wise measures for the fully-connected network trained on MNIST dataset.

