# OpenReview forum: "Feature-Robustness, Flatness and Generalization Error for Deep Neural Networks"
_ICLR.cc/2020/Conference — Reject_

### Official Review · AnonReviewer1 · 2019-10-23
**Official Blind Review #1**

**Rating:** 3

**Review:**

This paper proposes a flatness measure that is invariant to layer-wise reparametrizations in ReLU networks. The notion of feature robustness, which is a notion the paper proposes, connects the flatness measure to generalization error.

This paper should be rejected because it is not well-placed in the literature. Similar notions with the proposed flatness measure have repeatedly appeared in the literature. This paper needs to discuss novel insights.

Major comments:
1) Many studies proposed or mentioned the flatness measures listed in Table 1 [1, 2, 3, 4]. One of the most relevant work will be [1]. It appears that the Fisher-Rao norm [1] has several advantages over the proposed measure in the submitted paper.
A) Fisher-Rao norm is invariant to a broader range of linear transformations.
B) Fisher-Rao norm does not rely on the Hessian, which is more suitable for non-smooth ReLU networks.
Additionally and importantly, the Fisher-Rao norm has a direct connection with the size of input gradients, which has a strong relationship with the feature robustness. It is strongly encouraged to discuss the connections and comparisons with the Fisher-Rao norm.
2) On the connection to the generalization error, Theorem 10 relies on the strong assumption defined in Definition 9. Given the high ability of deep networks to express many functions, assuming that \phi(S) is epsilon-representative seems difficult to justify. This paper should discuss why the assumption is reasonable. Otherwise, it is hard to claim that this paper connected the modified flatness measure to generalization error.

[1] Liang et al. "Fisher-Rao Metric, Geometry, and Complexity of Neural Networks." AISTATS 2019
[2] Achille et al. "Emergence of Invariance and Disentanglement in Deep Representations." JMLR 19 (2018)
[3] Neyshabur et al. "Exploring Generalization in Deep Learning." NeurIPS 2017
[4] Tsuzuku et al. "Normalized Flat Minima: Exploring Scale Invariant Definition of Flat Minima for Neural Networks using PAC-Bayesian Analysis." arXiv:1901.04653

=====

Update:

Thank you for the replies and the clarifications. They did address some of my concerns. However, the theoretical result is limited, and I do not think it provides clear connections of flatness and the local loss landscape. I think this paper is not ready for publication, and I keep my score.

**Experience Assessment:**

I have read many papers in this area.

**Review Assessment: Checking Correctness Of Derivations And Theory:**

I assessed the sensibility of the derivations and theory.

**Review Assessment: Checking Correctness Of Experiments:**

I did not assess the experiments.

**Review Assessment: Thoroughness In Paper Reading:**

I read the paper at least twice and used my best judgement in assessing the paper.

---

> ### Author Response · Authors · 2019-11-12
> **specific reply to Reviewer 1**
>
> We thank the reviewer for directing us to related work. We have integrated references in a new version. Please see our general reply to all reviewers for a general discussion. Here, we want to point out the differences to the suggested work more explicitly:
>
> The Fisher-Rao norm is not a good measure of flatness at local minima as it only measures flatness into one specific direction. It corresponds to checking how the loss changes when we perturb the weights into the direction given by w only, no other perturbations are considered by this notion. In contrast, our measure takes the full spectrum of the Hessian into account. In the appendix we discuss how a variant of our proposed measure becomes also invariant under node-wise reparameterizations.
> Similarly, the expected sharpness of Neyshabur et al. is not a local measure of flatness.
>
> Tsuzuku et al.’s measure is very similar to ours and also invariant under Dinh’s reparameterizations. We were unaware of this preprint and now refer to it in our new version. However, we still argue that an approach different to PAC-Bayes bounds might be needed to connect the empirically observed relation between flatness and generalization: Their derivation (like ours) uses a second order Taylor approximation, which is usually valid only for a local neighborhood around the minimum, while it is used in a larger neighborhood of a size dependent on the weights and the Hessian. By localizing the priors to allow using the Taylor approximation, the KL-divergence term goes to infinity, though. In Section 5 we added a discussion why we consider the use of localized distributions in PAC-Bayesian bounds as problematic. As flatness and feature robustness are local measures, we argue that a different approach seems necessary.
> Achille et al. do not suggest or discuss measures of flatness when relating flatness to the amount of information in weights. However, it is true that our tracial flatness measure can be implicitly found in their Proposition 4.3. We think that this actually speaks for our proposed measure derived from feature robustness.
>
> We discuss a way to obtain a computable bound to our notion of epsilon-representativeness in appendix D. This bound uses a similarity between our notion of representativeness and kernel density estimation. With this, results on the error of KDE can be used to bound the representativeness. The required similarity only holds for distributions that are different to the ones used in Theorem 10, so that the generalization bound does not follow. However, it shows that our notion can be reasonable. Furthermore, the result in Appendix D  shows that epsilon-representativeness can be bounded independent of the model.

---

> > ### Comment · AnonReviewer1 · 2019-11-14
> > **Response**
> >
> > Thank you for your detailed response.
> >
> > Comments to specific reply:
> > 3) I understood that the proposed measure has a more direct connection to flatness compared to the Fisher-Rao norm. However, in terms of the direct link to flatness, the expected sharpness of Neyshabur et al. is more natural to the original motivation of flatness discussion. I do not understand why the flatness measure should be local. Please clarify this point. Concerning Tsuzuku et al., I would rather think that the preprint tries to address the KL-term issue. However, it is a mere recent preprint, and more comparisons with the preprint will not affect my score.
> > 4) Thank you for adding more explanation. However, I still do not think that the assumption in Theorem 10 is reasonable. Even if we can calculate an upper-bound of the epsilon-representativeness, if it becomes dominant, the proposed epsilon-robustness will not be a good measure of generalization. Authors are also responsible for explaining why the epsilon-representativeness should be small in deep networks.
> >
> > Comments to general reply:
> > 5) I understood that the submission has an advantage over PAC-Bayesian work such as Dziugaite and Roy or Neyshabur et al. that it potentially requires fewer assumptions. However, I think the advantage becomes essential only when the submission also provides tractable generalization bounds.
> >
> > Overall, I agree that the view in the paper is potentially impactful. However, the current submission seems to require several steps to connect the proposed notion to the flatness. I think this submission is still not ready for publication, and I keep my score.

---

> > > ### Author Response · Authors · 2019-11-15
> > > **Clarification on locality of flatness**
> > >
> > > Thank you for your answer and your interest, we value the discussion. We did not mean to claim that a suitable flatness measure must be local. Only Hessian-based measures of flatness must be local. It is our understanding of the results from literature that Hessian based measures are related to generalization, but it is unclear why. We show how Hessian-based flatness measures can be made invariant of reparameterizations, taking away the main argument speaking against them. We additionally relate Hessian-based flatness to a natural notion of feature robustness, which gives more evidence for Hessian-based measures. We agree that this does not yet explain the correlation to generalization. Therefore, we additionally display that there is indeed a promising approach to connect flatness to generalization through feature robustness. As a conclusion, a flatness measure must not be local, but the most intuitive and promising Hessian-based ones are.

---

### Official Review · AnonReviewer3 · 2019-10-26
**Official Blind Review #3**

**Rating:** 1

**Review:**

This paper proposes a notion of feature robustness which is invariant with respect to rescaling the weight. The authors discuss the relationship of this notion to generalization.

The definition of feature robustness is interesting and could potentially be useful. However, the paper has the following major issues:

1- Related work: It seems that authors are unaware of the related work in this area. There are many relevant work in this area that connect feature or weight robustness to generalization look at [1,2,3,4] for some examples. I suggest authors to do a comprehensive literature review.

2- Theoretical results: The theoretical results presented in the paper have very limited value. For example, authors fail to really connect their suggested measure to generalization in any meaningful way. Instead they end up decomposing the test error to the sum of their robustness measure and the gap between robustness and test error which is trivial. I suggest authors to look at the literature on PAC-Bayesian and compression-based bounds to connect their suggested measure to generalization.

3- Experiments: The experiments are not really convincing. The empirical results show that the suggested measure can correlate with generalization when training with different batch-sizes. When varying other things, the measure is not really correlated. Therefore, this is not any better than the version suggested by Keskar et. al. Moreover, the experiments are very limited and I suggest authors to look at more controlled setting to verify the relationship of their measure to generalization. Also, when looking at the generalization, it is important to set the stopping criterion based on the cross-entropy instead of number of epochs.

[1] Dziugaite and Roy. "Computing nonvacuous generalization bounds for deep (stochastic) neural networks with many more parameters than training data". AAAI, 2017.
[2] Neyshabur et. al. "Exploring Generalization in Deep Learning", NeurIPS 2017.
[3] Arora et. al. "Stronger generalization bounds for deep nets via a compression approach". ICML 2018.
[4] Wei and Ma. "Data-dependent Sample Complexity of Deep Neural Networks via Lipschitz Augmentation", NeurIPS 2019.

****************************
After author rebuttals:

Author have added discussion of related work which was missing in the original submission (thanks!). However, the other two issues are still present. On the theoretical side, I think the major issue is that the paper cannot connect the measure to generalization properly and ends up decomposing the test error to the sum of the robustness measure and the gap between test error and the robustness measure which is not informative. However, this would have been still interesting if the measure could go beyond other empirical measures. Unfortunately, the correlation only happens in case of changing batch size (and learning rate which we know is empirically equivalent to changing batch size) and therefore cannot go beyond what is shown in Keskar et al. 2017. Therefore, my evaluation remains the same.




**Experience Assessment:**

I have published in this field for several years.

**Review Assessment: Checking Correctness Of Derivations And Theory:**

I assessed the sensibility of the derivations and theory.

**Review Assessment: Checking Correctness Of Experiments:**

I assessed the sensibility of the experiments.

**Review Assessment: Thoroughness In Paper Reading:**

I read the paper at least twice and used my best judgement in assessing the paper.

---

> ### Author Response · Authors · 2019-11-12
> **specific reply to Reviewer 3**
>
> We thank the reviewer for directing us to related work. We have integrated the references in a new version. We note that all suggested papers study the PAC Baysian approach and derive generalization guarantees for averages over models (or in the case of Arora et al., a bound on the compression instead of the original function). Please see our general reply to all reviewers and the new content in Section 5 that argues for the necessity of a differing approach.
>
> Experiments: Our measure correlates with changing both batch size and learning rate, while changing initialization seems not to affect the generalization gap. Different to the measure proposed by Keskar, our measure is invariant with respect to reparameterizations, which is a crucial improvement. Regarding the stopping criterion, we considered that checking the stabilization of gradients (that is equivalent to stabilization in loss) is the most valuable criterion. The number of epochs was based on the stagnation period of loss changes, which explains our choice.

---

> > ### Comment · AnonReviewer3 · 2019-11-13
> > **Thanks for the response**
> >
> > Thank you for your response.
> >
> > Author have added discussion of related work which was missing in the original submission (thanks!). However, the other two issues are still present. On the theoretical side, I think the major issue is that the paper cannot connect the measure to generalization properly and ends up decomposing the test error to the sum of the robustness measure and the gap between test error and the robustness measure which is not informative. However, this would have been still interesting if the measure could go beyond other empirical measures. Unfortunately, the correlation only happens in case of changing batch size (and learning rate which we know is empirically equivalent to changing batch size) and therefore cannot go beyond what is shown in Keskar et al. 2017. Therefore, my evaluation remains the same.

---

### Official Review · AnonReviewer2 · 2019-10-27
**Official Blind Review #2**

**Rating:** 3

**Review:**

This paper describes a connection between flatness of minima and generalization in deep neural networks. The authors define a concept called "feature-robustness" and show that it is related to flatness. This is derived through a straightforward observation that perturbations in feature space can be recast as perturbations of the model in parameter space. This allows the authors to define a (layerwise) flatness measure for minima in deep networks (this layerwise flatness measure is also invariant to rescalings of the layers in neural networks with positively homogenous activations). The authors combine their notion of feature robustness with epsilon representativeness of a function to connect flatness to generalization. They present a few empirical evaluations on CIFAR10 and MNIST.

I believe this paper is able to once again confirm the relationship between flatness and generalization in an empirical manner with their layerwise measure of flatness. I am not so convinced about the theoretical justification that they claim to provide and thus do not recommend acceptance.

Theory - The key theorem relating generalization and flatness is Theorem 10 which says that if a compositional model is feature robust and the output of the first component is an epsilon-representative for the second component, then the compositional model will generalize. While this is interesting, it is not clear to me that this guarantees generalization for deep neural networks. This result only talks about feature robustness and representativeness for a particular layer. If a deep network has many layers, will the feature robustness layers closer to the input guarantee feature robustness at deeper layers? That might require a further unit operator norm constraint on the layer operator, which is a restriction on the types of weights that can be used. If a sample is epsilon representative at one layer, what is required for the next layer to be epsilon representative for the rest of the deep network? This seems to be a missing step in relating flatness/feature robustness of a layer to the generalization of the whole network.

Another idea that I think arises from Theorem 10 is that the flatness of loss landscapes is important when you have learning problems where the hypothesis class is compositional. While flatness is only spoken of in the case of deep neural networks, can we identify the same phenomenon in other problems? I would encourage the authors to try and identify another model in which the flatness-generalization relationship exists (even empirical evidence would suffice for now). This would strengthen the case for studying flatness and biasing optimization towards flatter solutions in the case of deep networks.

Experiments - This section seems to be pretty rudimentary, I would like to see more results on different kinds of network architectures (VGG? Inception? AlexNet?), more datasets (KMNIST? Fashion MNIST? SVHN?), and possibly more repetitions. At one point the authors mention that they declare a minimum has been reached if the training loss is < 0.07. Atleast on CIFAR10 and MNIST it is possible to achieve training loss <1e-4 so am not sure if the networks that the authors are testing are minima at all (It is important for them to be minima since the flatness measure is only defined at minima). Can the authors also identify more situations other than large batch vs small batch training that would lead them to obtain flatter/sharper minima?

The authors also claim that measuring generalization using test error is flawed, but do not provide details about their method of measuring generalization. I would want to see these details and a more thorough discussion of why measuring generalization through test error is flawed.

While this is an interesting paper, I do not believe it is ready for acceptance at ICLR 2020.



**Experience Assessment:**

I have published one or two papers in this area.

**Review Assessment: Checking Correctness Of Derivations And Theory:**

I carefully checked the derivations and theory.

**Review Assessment: Checking Correctness Of Experiments:**

I carefully checked the experiments.

**Review Assessment: Thoroughness In Paper Reading:**

I read the paper thoroughly.

---

> ### Author Response · Authors · 2019-11-12
> **specific reply to Reviewer 2**
>
> We thank the reviewer for his good ideas and suggestions to strengthen our results. Please see our general reply to all reviewers for a general discussion of these results.
>
> We note that our approach suggests that it suffices to achieve feature robustness in one layer. Further operator norm constraints of later layers are implicitly encoded in the feature robustness.
>
> The suggestion to apply our theory to other decomposable models is a very interesting idea which we will follow up on in future work.
>
> Regarding the experiments, we would like to remark that empirical proof of the generalization-flatness correlation was not the goal of our paper. Since there has been several experiments in existing literature that connect flatness to feature robustness, it was only our experimental goal to empirically validate that our correction factor does not destroy the correlation. Since we achieved this by our empirical results, we considered the results as sufficient. However, we acknowledge that more experiments would strengthen our case and will include them in a later version of the paper.
>
> The networks on MNIST were learned by a simple fully connected network, which was not used in any known experiments, so 0.07 was found empirically through multiple runs till gradients were stagnating. Our experiments include different initializations, different batch size, and learning rate.
> Regarding the problem of test accuracy, our point was that training accuracies often do not reach one, so only looking at test accuracies can be misleading when studying generalization.

---

### Public Comment · ~Micah_Goldblum1 · 2019-11-08
**An Interesting Connection**

Hi Authors,
Thank you for your interesting paper.  I noticed that your work concerning generalization is related to our paper which visualizes the sharp-flat phenomenon, including experiments on realistic architectures, as an explanation for generalization.[1]  Please consider mentioning the relationship with our work in your next version.

[1] Huang, W. Ronny, et al. "Understanding Generalization through Visualizations." arXiv preprint arXiv:1906.03291 (2019).

---

### Author Response · Authors · 2019-11-12
**To all Reviewers**

We thank the reviewers for their comments and hints to the related literature which we integrated into our paper. However, this does not reduce our contribution: Our paper derives a measure of flatness naturally from a suitable notion of feature robustness. In contrast to previously published work, the measure takes the whole spectrum of the Hessian into account-and therefore qualifies as a measure of flatness-and is invariant to reparameterization. Flatness and Hessian based measures of flatness have been correlated to generalization several times in existing literature. Dinh et al.’s work raises the question, why these measures of flatness based on the spectrum of the Hessian seem to be related to generalization. Our natural modification frees Hessian-based measures from their "Dinh et al."-curse. With this, we bring the natural flatness measures that consider the full spectrum of the Hessian back into the game. This has not been achieved in previously published work (except for possibly the preprint [1] we were not aware of during writing). The Fisher-Rao Norm [2] does not consider the full spectrum of the Hessian. Therefore, this measure does not describe the flatness of the loss surface in arbitrary directions. But existing literature contains a lot of empirical support that Hessian-based flatness is related to generalization. To support our theory, we argue that we only need to show that our correction factor does not destroy this correlation, which we do empirically.

The connection of our flatness measure to feature robustness provides a mean to theoretically analyze generalization. Remarkably, this theory does not require a PAC-Bayesian approach, but instead uses the natural and local definition of feature robustness. Our approach fundamentally differs from PAC-Bayes guarantees that bound the expected error of a hypothesis drawn from a learned posterior. Moreover, PAC-Bayesian bounds are often valid only for the 0-1 loss in classification. Our theory is independent of the loss function (as long as it is twice differentiable) and independent of the prediction task. Given this difference, we missed to refer to the suggested work on PAC-Bayesian bounds but argue analytically to reach similar (but novel) conclusions, which we consider remarkable. We understand, however, that this resulted in a lack of acknowledgement of part of the community. In particular, since related work has (as we now see) appeared in 2019 under the PAC-Bayesian framework. We agree that the 2019 papers and preprints are essential to mention. We integrated the recommended work.

Our paper tackles generalization from an angle that has not been considered, yet. We argue that this particular view on the topic is not only interesting, but possibly even necessary. Feature robustness and flatness of the loss surface are purely local properties. To study such local properties in a PAC-Bayesian bound requires strongly localized posterior distributions. But with a data-independent prior distribution, the Kullback Leibler divergence between prior and posterior diverges to infinity with increasing localization. We added content on related PAC-Bayesian approaches and the problems that arise when using localized measures. While we do not give a full theory to connect our notions to generalization in terms of sharp generalization bounds, our work discusses how such a theoretical connection must incorporate a measure of representativeness to connect a local property like flatness or feature robustness to a global property on the entire (unknown) distribution. We are hopeful that this connection can be tightened in the future for specific losses. Without question, it suggests a new approach to generalization bounds independent of the chosen loss function, learning algorithm, and hypothesis class.

---

> ### Author Response · Authors · 2019-11-12
> **... ctd**
>
> We also want to stress again that our theoretical framework makes very little assumptions: the model has to be decomposable and at a local optimum. Moreover, all involved functions need to be twice differentiable. We make no further assumptions on the loss function (e.g., Lipschitz, convexity, or smoothness (see Chp. 12.1 in [3]), on the hypothesis class (e.g., VC dimension [4] or Rademacher complexity [5]), the prediction task, or on the learning algorithm (e.g., stability, or that it is an empirical risk minimizer, or uniform convergence property).
>
> The definition of feature robustness in our paper differs from classical robustness (e.g., [6]) in that it is a local measure on a given dataset. This allows to compute it (or rather bound it) on a given dataset using a measure of flatness - which, conveniently, is also invariant to certain reparameterizations. Moreover, feature robustness is measured for a particular feature representation of a model, e.g., a particular layer in a neural network. The guarantees that follow from that hold for the entire model - it suffices that only one layer is feature robust. Lastly, feature robustness is a property of a model given a dataset and as such independent of the learning algorithm and training dataset. Given that the data distribution is benign (as measured by the representativeness of the input data), feature robustness implies good generalization capabilities. This bound is not distribution independent, like classical PAC bounds, but should allow for tighter guarantees, since it takes into account the hardness of the particular learning problem (one indication for this is that the novel definition of representativeness is always smaller or equal to classical representativeness).
>
> We admit that, since we were not able to provide a directly computable bound, yet, we cannot quantify the advantage of this leverage. Finding such a connection would lead to novel insights into the generalization capabilities of neural networks and could provide an entirely new view on statistical learning theory. This could close a fundamental gap between the current theoretical understanding of neural networks and their practical success.
>
>
> [1] Tsuzuku et al. "Normalized Flat Minima: Exploring Scale Invariant Definition of Flat Minima for Neural Networks using PAC-Bayesian Analysis." arXiv:1901.04653
> [2] Liang et al. "Fisher-Rao Metric, Geometry, and Complexity of Neural Networks." AISTATS 2019
> [3] Shalev-Shwartz, Shai, and Shai Ben-David. Understanding machine learning: From theory to algorithms. Cambridge university press, 2014.
> [4] Vapnik, Vladimir, Esther Levin, and Yann Le Cun. "Measuring the VC-dimension of a learning machine." Neural computation 6.5 (1994): 851-876.
> [5] Bartlett, Peter L., Olivier Bousquet, and Shahar Mendelson. "Local rademacher complexities." The Annals of Statistics 33.4 (2005): 1497-1537.
> [6] Huan Xu and Shie Mannor. Robustness and generalization. Machine learning, 86(3):391–423, 2012

---

### Decision · Program_Chairs · 2019-12-19

**Decision:**

Reject

**Comment:**

The authors propose a notion of feature robustness, provide a straightforward decomposition of risk in terms of this robustness measure, and then provide some empirical evidence for their perspective. Across the board, the reviewers raised issues with missing related work, which the authors then addressed. I will point out that some things the authors say about PAC-Bayes are false. E.g., in the rebuttal the authors say that PAC-Bayes is limited to 0-1 error. It is generally trivial to obtain bounds for bounded loss. For unbounded loss functions, there are bounds based on, e.g., sub gaussian assumptions.

Despite improvements in connections with related work, reviewers continued to find the theoretical contributions to be marginal. Even the empirical contributions were found to be marginal.

---

> ### Author Response · Authors · 2020-02-10
> **Clarification from the authors**
>
> We would like to clarifiy that we did not claim that PAC-Bayes would be limited to the 0-1 error. We only remarked that bounds are often derived for the 0-1 error. In particular, the related literature suggested by reviewers considers the 0-1 error.